# Sex differences in metabolic pathways are regulated by *Pfkfb3* and *Pdk4* expression in rodent muscle

Antonius Christianto[1], Takashi Baba [1,2 ✉], Fumiya Takahashi[1], Kai Inui[1], Miki Inoue[1,2], Mikita Suyama[3], Yusuke Ono[4,5], Yasuyuki Ohkawa[6] & Ken-ichirou Morohashi [1,2]

Skeletal muscles display sexually dimorphic features. Biochemically, glycolysis and fatty acid β-oxidation occur preferentially in the muscles of males and females, respectively. However, the mechanisms of the selective utilization of these fuels remains elusive. Here, we obtain transcriptomes from quadriceps type IIB fibers of untreated, gonadectomized, and sex steroid-treated mice of both sexes. Analyses of the transcriptomes unveil two genes, *Pfkfb3* (*phosphofructokinase-2*) and *Pdk4* (*pyruvate dehydrogenase kinase 4*), that may function as switches between the two sexually dimorphic metabolic pathways. Interestingly, *Pfkfb3* and *Pdk4* show male-enriched and estradiol-enhanced expression, respectively. Moreover, the contribution of these genes to sexually dimorphic metabolism is demonstrated by knockdown studies with cultured type IIB muscle fibers. Considering that skeletal muscles as a whole are the largest energy-consuming organs, our results provide insights into energy metabolism in the two sexes, during the estrus cycle in women, and under pathological conditions involving skeletal muscles.

[1] Division of Molecular Life Science, Graduate School of Systems Life Science, Kyushu University, Maidashi 3-1-1Higashi-ku Fukuoka 812-8582, Japan.
[2] Department of Molecular Biology, Graduate School of Medical Sciences, Kyushu University, Maidashi 3-1-1Higashi-ku Fukuoka 812-8582, Japan. [3] Division of Bioinformatics, Medical Institute of Bioregulation, Kyushu University, Maidashi 3-1-1Higashi-ku Fukuoka 812-8582, Japan. [4] Department of Muscle Development and Regeneration, Institute of Molecular Embryology and Genetics, Kumamoto University, Honjo 2-1-1Chuo-ku Kumamoto 860-0811, Japan. [5] Center for Metabolic Regulation of Healthy Aging, Kumamoto University Faculty of Life Sciences, Honjo 1-1-1Chuo-ku Kumamoto 860-8556, Japan. [6] Division of Transcriptomics, Medical Institute of Bioregulation, Kyushu University, Maidashi 3-1-1Higashi-ku Fukuoka 812-8582, Japan.
✉email: takbaba@cell.med.kyushu-u.ac.jp

Animal species have developed a variety of sex differences in their structures and functions. In most animals, this sexual dimorphism is most obvious in the reproductive system. Additionally, however, many animals exhibit sexually dimorphic appearances involving body size, exterior body parts, and feather color. Even many mammalian species demonstrate visually appealing features that differ between males and females. Skeletal muscle is a representative organ whose structures and activities differ between the two sexes[1].

Skeletal muscle is composed of multiple types of fibers that differ in terms of their morphological, biochemical, and physiological properties. In rodents, these fibers are largely divided into four types (types I, IIA, IIB, and IIX) based on which the myosin heavy chain (*MYH*) gene is expressed[2–4]. As for the functional features of the fibers, type I demonstrates the slowest contraction, while types IIA, IIX, and IIB exhibit successively faster contraction[5,6]. Morphologically, the type IIB fiber is the largest and the type I fiber is the smallest. Biochemically, type IIB fiber has the highest glycolytic activity, while type I fiber demonstrates the highest oxidative capacity. These differences in energy metabolism and the differential ATPase activities of myosin heavy chains have been studied in relation to functional differences among the fiber types[1,7]. In addition to these representative fibers, other fiber types that express multiple types of MYH or minor types of MYH have been detected[8,9].

The sexually dimorphic features of skeletal muscles have been investigated extensively. The total mass of muscle fibers and their individual sizes, fiber type composition, and skeletal muscle energy metabolism, contractile strength, and fatigability were found to be different between the two sexes[1,10,11]. In addition, the preferred fuels for energy metabolism have been shown to be glucose and fatty acids in the muscles of males and females, respectively[12,13]. Although this difference has been thought to play a fundamental role in the sexually dimorphic functions of skeletal muscle, the mechanisms that induce this feature remain unknown. In addition to these morphological, physiological, and biochemical studies, recent deep-sequence studies revealed sexually dimorphic gene expression in skeletal muscles[14–17].

Sex steroids have been investigated as the primary mechanism underlying sex differences[1,18,19]. In fact, some male-specific characteristics of skeletal muscles, such as heavier muscle weight and larger fiber size, were shown to be the result of testosterone[20,21]. Testosterone acts by binding to the androgen receptor (AR/NR3C4), thereby regulating target gene expression. Several genes whose functions are closely related to the male-biased anabolic activity of muscles were shown to be targets of AR[22–24].

Many studies have advanced our understanding of the functional characteristics of skeletal muscles. Unfortunately, however, much of this research was performed with whole muscle rather than with specific fiber types. Since skeletal muscle consists of multiple fiber types, investigation of each type is thought to be essential for the overall comprehension of the functional properties of skeletal muscle. Therefore, in the present study, we aimed to investigate sexual dimorphisms of skeletal muscle at the level of particular fiber types. Our analyses of transcriptome datasets revealed that two key genes underlie sexually dimorphic metabolism. The male-predominant glycolytic activity was found to be due to male-enriched expression of *Pfkfb3* (phosphofructokinase-2), while female-predominant fatty acid β-oxidation was found to be due to E2 (estradiol)-enhanced expression of *Pdk4* (pyruvate dehydrogenase kinase 4).

## Results

**Sexual dimorphism in muscle fiber size**. We attempted to confirm sexual dimorphism in muscle fiber size (cross-sectional area (CSA)) in five different skeletal muscles (quadriceps, tibialis anterior, gastrocnemius, triceps, and soleus) in 8-week-old male and female mice. In all skeletal muscles examined, fiber CSAs were larger in males than in females (Supplementary Fig. 1a). Mammalian skeletal muscles consist of four types classified according to the type of myosin heavy chain expressed: type I, IIA, IIB, and IIX fibers can be distinguished by the predominant expression of *MYH7* (encoding MYH1), *MYH2* (encoding MYH2A), *MYH4* (encoding MYH2B), and *MYH1* (encoding MYH2X), respectively[4,25,26]. Antibodies against MYH2A and MYH2B were used to identify type IIA and IIB fibers, respectively (Supplementary Fig. 1b). In the skeletal muscles examined, the CSAs of type IIB fibers (a major fast fiber type in fast-twitch skeletal muscles in rodents) were larger in males than in females. We focused on this fiber type in the following studies to further investigate the sexual dimorphism in CSA.

We next addressed the age at which the sex difference emerged. The CSAs of quadriceps type IIB fibers were examined postnatally at 2, 3, 4, and 8 weeks. A slight difference between sexes appeared at 4 weeks, and the difference became obvious at 8 weeks (Supplementary Fig. 1c). *Amd* (S-adenosylmethionine decarboxylase) and *Smox* (spermine oxidase), both of which are required for polyamine synthesis, were reported to exhibit male-enriched and androgen-induced expression[14,15,17,27]. The expressions of these genes were examined in quadriceps type IIB fibers. Similar to the CSAs, slight male-enriched expressions were observed at 4 weeks, and were clearly apparent at 8 weeks (Supplementary Fig. 1d).

**Effect of sex steroids on muscle fiber size**. To examine the effect of sex steroids on CSAs, we prepared muscles from sham-operated males and females, castrated males (Cas), ovariectomized females (Ovx), mice treated with DHT (dihydrotestosterone) after gonadectomy (Cas+DHT and Ovx+DHT), and mice treated with E2 after gonadectomy (Cas+E2 and Ovx+E2), as shown in Supplementary Fig. 2. Of note, as the control female, we used diestrus mice that had undergone sham operation.

After the skeletal muscles of the mice were stained for MYH2B (Fig. 1a), the CSAs of the MYH2B-positive type IIB fibers were measured (Fig. 1b and Supplementary Fig. 1e). Immunofluorescence analysis suggested that type IIB fibers were larger in males than in females, and that DHT treatment enlarged CSAs regardless of sex. Statistical analyses indicated that the CSAs of the quadriceps, tibialis anterior, triceps, and soleus were larger in males than in females (Fig. 1c). With a few exceptions, DHT increased the CSAs of the muscles above regardless of sex, whereas E2 did not. Interestingly, the quadriceps fibers enlarged by DHT remained larger in males than in females.

**Sexually dimorphic gene expression in quadriceps type IIB fibers**. Since the CSAs of quadriceps muscle type IIB fibers exhibited clear sexual dimorphism and DHT dependency, we decided to obtain transcriptomes of these fibers. In all 10 experimental mouse groups (sham-operated males and females, males and females transplanted with a DHT-containing or empty pellet after gonadectomy, and males and females injected with E2-containing or corn oil after gonadectomy), single fibers were prepared from an area of the quadriceps where ~95% fibers are type IIB (Supplementary Fig. 3a). Thereafter, the fiber types were determined by RT-PCR (Supplementary Fig. 3b).

The RNAs recovered from fibers positive for MYH2B (encoded by *Myh4*) were pooled and subjected to mRNA sequencing. Transcriptome datasets with sufficient quality for the following analyses were obtained from all experimental groups above (Supplementary Fig. 3c). Genes whose CPM (counts per million

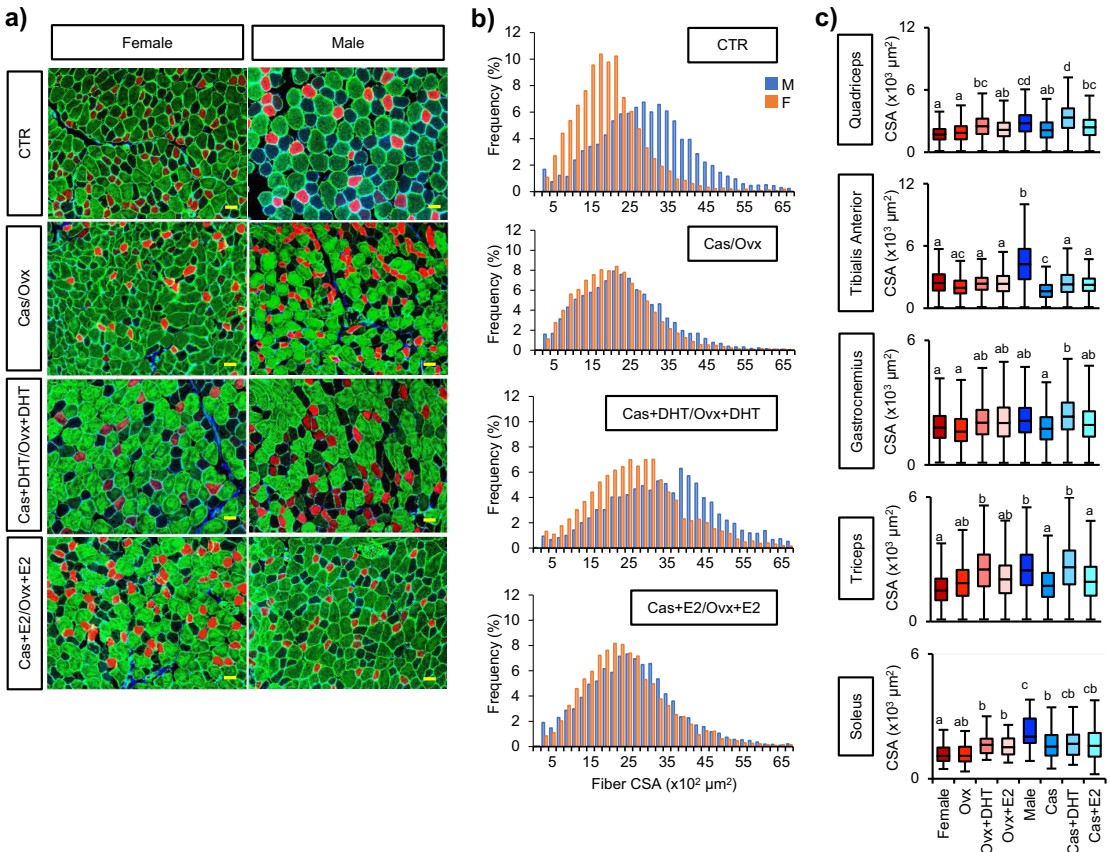

**Fig. 1 Effects of sex steroids on skeletal muscle fiber size. a** Quadriceps muscle samples were prepared from eight experimental groups: sham-operated male and female mice (CTR), gonadectomized mice (Cas for males and Ovx for females), DHT-treated male mice (Cas+DHT), and female mice (Ovx+DHT) after gonadectomy, and E2-treated male mice (Cas+E2) and female mice (Ovx+E2) after gonadectomy (Supplementary Fig. 2). Type IIA (red) and type IIB (green) fibers were detected with MYH2A and MYH2B antibodies, respectively. Blue: DAPI. Scale bar = 50 μm. **b** The CSAs of the type IIB fibers were measured using approximately 4800 to 6000 fibers per specimen. The distributions of CSA sizes (horizontal axis) and frequencies (vertical axis) are shown. Male and female data are indicated by blue and orange bars, respectively. The same studies were performed using the tibialis anterior, gastrocnemius, triceps, and soleus muscles (Supplementary Fig. 1e). A representative result from three biologically independent samples is shown. **c** The CSA size distribution was compared among the eight experimental groups above. The data were analyzed as described in the "Methods" section. The box and whisker plots with the same letter are not significantly different from each other ($p < 0.01$).

mapped reads) values were >10.0 in either the sham-operated males or females were extracted as all expressed genes (6978 genes) and utilized for the following analyses. Of these genes, 68 and 60 demonstrated more than 2.0-fold enrichment in males and females, respectively (Fig. 2a, Table 1). As described in detail below, two key genes for energy metabolism, *Pdk4* (pyruvate dehydrogenase kinase 4) and *Pfkfb3* (phosphofructokinase-2), were among the male-enriched genes.

Expression profiles of the male- and female-enriched genes in the 10 experimental groups were analyzed by hierarchical clustering (Fig. 2b, c). According to the clustering profile of the male-enriched genes, male, Cas+DHT, and Ovx+DHT mice were classified into one subgroup. Nearly half of the male-enriched genes exhibited DHT-dependent expression. A similar effect of DHT was observed in the ovariectomized females. Likewise, clustering of the female-enriched genes indicated that the same experimental groups were likely to form a subgroup. Comparison of female, Cas+E2, and Ovx+E2 mice suggested that a certain number of the female-enriched genes were activated by E2 in both sexes. Even though both DHT and E2 affected gene expression, it is likely that the effects of DHT are more evident than those of E2.

We unexpectedly found that a group of male-enriched genes was activated in castrated males (Cas+P) following empty pellet

implantation, although this phenomenon was not observed in ovariectomized females (Ovx+P). Since the reason for this unexpected gene activation by control treatment was unknown, we carefully examined the subsequent results.

Principal component analysis of whole transcriptome data was used to classify the mouse groups. Male, Cas+DHT, and Ovx +DHT mice comprised a separate subgroup (SG1 in Fig. 2d). The classification of female, Cas+E2, and Ovx+E2 mice was unclear. The experimental group, Cas+P, was classified apart from the others, perhaps due to the unexpected gene activation by the empty pellet implantation described above. It was reported that testosterone levels were different among mouse strains, and that of C57BL/6 was significantly lower than those of CD-1, CH3, and FVB[28]. Studies using mice with higher testosterone levels might provide us with more pronounced sexually dimorphic gene expression.

**Functions related to male- and female-enriched genes.** Gene ontology analyses were conducted on the sex-biased genes. The polyamine biosynthetic process was identified as a potential biological process related to the male-enriched genes (Supplementary Table 1). Among the polyamine synthetic genes, *Odc1* (ornithine decarboxylase 1), *Amd1/2*, and *Smox* showed higher expressions in males than in females and were induced by DHT

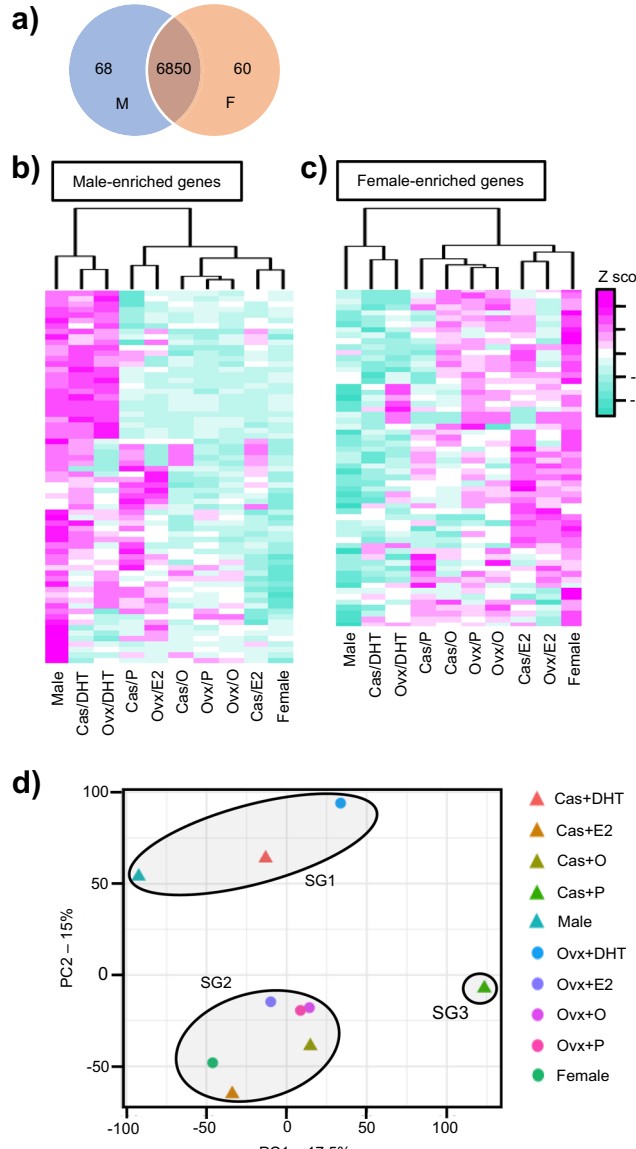

**Fig. 2 Sexually dimorphic gene expression in quadriceps type IIB fibers. a** A total of 6978 genes whose CPM values were more than 10.0 in either of the two sexes (sham-operated males or females) were analyzed. Overall, 68 and 60 genes were found to be male enriched (M) and female enriched (F) by more than 2.0-fold, respectively. **b**, **c** Heatmaps of the male-enriched and female-enriched gene expressions in the 10 experimental mouse groups (see text and Supplementary Fig. 2) are shown. Color gradients correspond to the z-score as indicated at the right. **d** Results of principal component analysis of whole gene expression in the 10 groups are shown. PC1 and PC2 account for 17.5% and 15% of the percentage contribution to the variance, respectively. As indicated by closed ovals, the 10 groups are divided into three subgroups (SG1–SG3).

treatment in both sexes (Supplementary Table 2a). The expressions of these genes were not upregulated by empty pellet implantation.

These polyamine synthetic genes were previously shown to be male enriched and androgen inducible[14,15,17,27]. As for their function in skeletal muscles, they were reported to suppress muscle atrophy and promote hypertrophy[29,30], possibly by regulating cellular proliferation and viability[31,32] as well as protein synthesis[33] and autophagy[34,35]. Together, it is likely that the sexual differences

in muscle sizes primarily depend on the amounts of androgens in males and females.

As for the female-enriched genes, the terms "collagen fibril organization," "wound healing," and "skeletal system development" were found by the gene ontology analysis. Since collagen genes were commonly included in these processes, we examined their expressions in the datasets. The expressions of many collagen genes were higher in females than in males (Supplementary Table 2b) but were not clearly affected by ovariectomy or E2 treatment.

**Differential regulation of *Pdk4* and *Pfkfb3* genes by sex steroids.** Because diestrus mice with low E2 serum concentrations were used as the control females, it was assumed that a certain population of genes potentially activated by E2 would not be included in the female-enriched genes. Therefore, the E2 induction ratios for all of the expressed genes (6978) were calculated in males and females (Cas+E2/Cas+oil and Ovx+E2/Ovx+oil) and plotted in Fig. 3a. The accumulation of genes in the upper right and lower left quadrants suggested that many genes were activated and suppressed by E2 regardless of sex. However, this pattern was not observed when only the female-enriched genes were analyzed (Fig. 3b). Moreover, many genes exhibiting relatively high activation or suppression by E2 were excluded from the female-enriched genes. Female-enriched *Col* genes were not activated intensively by E2, even in female mice. As a consequence, this analysis identified E2-induced genes that were not members of the female-enriched gene population. Interestingly, two key genes regulating energy metabolism, *Pdk4* and *Pcx* (pyruvate carboxylase), were included (Fig. 3a).

Likewise, the DHT induction ratios for all of the expressed genes and the male-enriched genes were calculated (Fig. 3c, d). Expectedly, *Smox* and *Amd1/2* were localized in the upper right quadrant, indicating that they were activated by DHT regardless of sex. Although *Pdk4* and *Pfkfb3* were male-enriched genes, their localizations differed from those of *Smox* and *Amd1/2*, suggesting that their expressions were relatively unaffected by DHT in both sexes.

**Possible contribution of PFKFB3 to male-predominant glycolysis.** PFKFB3 plays a crucial role in glycolytic regulation by producing fructose-2,6-bisphosphate, which robustly activates PFKM (the type of phosphofructokinase-1 found in muscles)[36,37] (Fig. 4a). Therefore, the male-enriched expression of *Pfkfb3* suggested that glycolytic activity in quadriceps type IIB fibers would be higher in males than in females. To examine this, we prepared muscle fibers from a particular region (white quadriceps)[38] of the quadriceps (Supplementary Fig. 3a), and the extracellular acidification rate (ECAR), an index of glycolytic activity, was determined. As expected, the ECAR was approximately two-fold higher for muscle fibers from males than from females (Fig. 4b).

We assumed that male-biased glycolytic activity could be caused by higher expressions of glycolytic genes in males as well as by the upregulated expression of *Pfkfb3*. However, neither the transcriptome data nor qRT-PCR showed male-enriched expression of any glycolytic genes (Fig. 4c). Moreover, their expressions were not largely affected by gonadectomy or sex steroid treatments. In contrast, both transcriptomic analysis and qRT-PCR (Fig. 4d, e) showed that *Pfkfb3* expression was decreased significantly by castration but not by ovariectomy. Consistent with the results shown in Fig. 3a and b, DHT treatment failed to reverse the decreased expression caused by castration. Empty pellet implantation did not affect the expression of *Pfkfb3*.

**Table 1 List of male-enriched (left) and female-enriched (right) genes.**

| Rank | Gene | CPM Male | CPM Female | FC[a] | Rank | Gene | CPM Female | CPM Male | FC[a] |
|---|---|---|---|---|---|---|---|---|---|
| 1 | Ddx3y | 73.69 | 0.00 | 7370.43 | 1 | Xist | 346.49 | 4.57 | 75.73 |
| 2 | Uty | 22.08 | 0.00 | 2208.98 | 2 | Tsix | 45.45 | 0.86 | 52.48 |
| 3 | Eif2s3y | 28.32 | 0.10 | 260.68 | 3 | Cyp4f39 | 22.20 | 3.42 | 6.47 |
| 4 | Kdm5d | 18.37 | 0.10 | 169.11 | 4 | Mki67 | 10.21 | 1.78 | 5.70 |
| 5 | Cyp17a1 | 11.70 | 0.20 | 56.47 | 5 | Mrc1 | 34.05 | 6.28 | 5.42 |
| 6 | Vaultrc5 | 18.12 | 0.89 | 20.19 | 6 | Fgfr4 | 10.51 | 2.35 | 4.45 |
| 7 | Cish | 42.27 | 2.52 | 16.73 | 7 | Rian | 59.06 | 13.77 | 4.29 |
| 8 | Tfcp2l1 | 22.47 | 1.73 | 12.94 | 8 | Igfn1 | 91.73 | 26.86 | 3.41 |
| 9 | 1700001O22Rik | 10.09 | 0.89 | 11.25 | 9 | Atp9a | 27.19 | 8.92 | 3.05 |
| 10 | Npnt | 13.45 | 1.38 | 9.67 | 10 | Meg3 | 69.62 | 23.33 | 2.98 |
| 11 | Amd1 | 1561.74 | 200.19 | 7.80 | 11 | Gramd1b | 36.27 | 12.56 | 2.89 |
| 12 | Amd2 | 1561.74 | 200.19 | 7.80 | 12 | Actc1 | 4141.90 | 1455.73 | 2.85 |
| 13 | Slc30a2 | 10.95 | 1.38 | 7.88 | 13 | Selenbp1 | 10.12 | 3.57 | 2.83 |
| 14 | Atp1b1 | 11.91 | 2.12 | 5.59 | 14 | Igf2 | 71.15 | 25.54 | 2.79 |
| 15 | Cdk19 | 59.85 | 11.15 | 5.36 | 15 | Peg3 | 75.05 | 27.89 | 2.69 |
| 16 | Igf1 | 59.43 | 11.15 | 5.33 | 16 | Col1a1 | 248.54 | 95.10 | 2.61 |
| 17 | 3000002C10Rik | 11.34 | 2.17 | 5.21 | 17 | Car3 | 3034.73 | 1176.83 | 2.58 |
| 18 | Pdk4 | 348.89 | 70.66 | 4.94 | 18 | Angpt1 | 10.95 | 4.28 | 2.56 |
| 19 | Mafb | 78.72 | 17.12 | 4.60 | 19 | Fbxl22 | 17.22 | 6.92 | 2.49 |
| 20 | Smox | 468.56 | 106.53 | 4.40 | 20 | Ces1d | 78.80 | 32.39 | 2.43 |
| 21 | Casp12 | 11.59 | 2.57 | 4.50 | 21 | Itga9 | 23.44 | 9.60 | 2.44 |
| 22 | Ptpn3 | 19.15 | 4.29 | 4.45 | 22 | Stbd1 | 84.72 | 34.85 | 2.43 |
| 23 | Chac1 | 73.02 | 17.27 | 4.23 | 23 | Aqp4 | 56.50 | 23.22 | 2.43 |
| 24 | Timp4 | 13.73 | 3.60 | 3.80 | 24 | Snx30 | 10.51 | 4.28 | 2.45 |
| 25 | Npc1 | 99.20 | 26.65 | 3.72 | 25 | Col3a1 | 386.11 | 159.66 | 2.42 |
| 26 | Tiam1 | 29.00 | 8.49 | 3.41 | 26 | Mxra8 | 23.98 | 9.95 | 2.41 |
| 27 | Cbr2 | 14.16 | 4.44 | 3.18 | 27 | Sfxn2 | 14.16 | 5.92 | 2.39 |
| 28 | Odc1 | 137.33 | 47.02 | 2.92 | 28 | Lrtm1 | 130.07 | 55.40 | 2.35 |
| 29 | Stab2 | 22.26 | 7.70 | 2.89 | 29 | Gm13031 | 10.16 | 4.32 | 2.35 |
| 30 | Rcan1 | 61.74 | 21.51 | 2.87 | 30 | Padi2 | 148.62 | 64.38 | 2.31 |
| 31 | Pfkfb3 | 1044.88 | 366.97 | 2.85 | 31 | Hist1h2be | 17.07 | 7.38 | 2.31 |
| 32 | Spon1 | 10.70 | 3.75 | 2.85 | 32 | Loxl1 | 11.74 | 5.07 | 2.32 |
| 33 | Rcan2 | 18.48 | 6.56 | 2.81 | 33 | Tceal7 | 25.56 | 11.13 | 2.30 |
| 34 | Jak3 | 12.09 | 4.34 | 2.78 | 34 | Lmcd1 | 31.73 | 13.88 | 2.29 |
| 35 | Lincpint | 27.4 | 10.0 | 2.75 | 35 | 6330410L21Rik | 13.03 | 5.74 | 2.27 |
| 36 | Pygo1 | 12.3 | 4.6 | 2.68 | 36 | Btg2 | 19.79 | 8.92 | 2.22 |
| 37 | Mettl21c | 83.5 | 32.3 | 2.59 | 37 | Rhobtb1 | 40.02 | 18.26 | 2.19 |
| 38 | Spns2 | 75.2 | 29.2 | 2.58 | 38 | H19 | 521.71 | 239.63 | 2.18 |
| 39 | Cited2 | 71.0 | 27.6 | 2.57 | 39 | Kcnc3 | 19.15 | 8.85 | 2.16 |
| 40 | Adarb1 | 12.4 | 4.9 | 2.54 | 40 | Myc | 10.21 | 4.71 | 2.17 |
| 41 | Mb | 30.7 | 12.2 | 2.52 | 41 | Col4a2 | 190.6 | 90.39 | 2.11 |
| 42 | Socs2 | 23.1 | 9.4 | 2.47 | 42 | Dpy19l3 | 12.39 | 5.81 | 2.13 |
| 43 | Lpl | 47.2 | 19.5 | 2.42 | 43 | Lgmn | 23.59 | 11.13 | 2.12 |
| 44 | Cd24a | 157.7 | 66.9 | 2.36 | 44 | Foxo6 | 20.18 | 9.52 | 2.12 |
| 45 | Fabp4 | 44.1 | 18.8 | 2.36 | 45 | Col18a1 | 11.60 | 5.46 | 2.12 |
| 46 | Serpinb6a | 137.4 | 59.3 | 2.32 | 46 | Kazald1 | 10.16 | 4.78 | 2.12 |
| 47 | Sh3d19 | 10.5 | 4.6 | 2.32 | 47 | Pnpla3 | 18.01 | 8.60 | 2.09 |
| 48 | Acsl3 | 59.6 | 26.3 | 2.28 | 48 | Mustn1 | 14.75 | 7.06 | 2.09 |
| 49 | Fhl1 | 19.1 | 8.4 | 2.28 | 49 | Itm2a | 181.34 | 87.46 | 2.07 |
| 50 | Msrb1 | 115.3 | 51.1 | 2.26 | 50 | B230312C02Rik | 106.19 | 51.26 | 2.07 |
| 51 | Slc43a3 | 11.9 | 5.3 | 2.28 | 51 | C130080G10Rik | 20.92 | 10.06 | 2.08 |
| 52 | Golm1 | 20.8 | 9.5 | 2.20 | 52 | St6galnac4 | 20.92 | 10.06 | 2.08 |
| 53 | Ar | 118.5 | 54.1 | 2.19 | 53 | Thbd | 21.12 | 10.27 | 2.05 |
| 54 | Cmtm6 | 10.4 | 4.7 | 2.21 | 54 | Gm5105 | 72.19 | 35.35 | 2.04 |
| 55 | Prkg1 | 27.4 | 12.5 | 2.19 | 55 | Obsl1 | 204.28 | 100.23 | 2.04 |
| 56 | St3gal5 | 23.2 | 10.7 | 2.18 | 56 | Tbc1d1 | 67.95 | 33.42 | 2.03 |
| 57 | Gadl1 | 18.4 | 8.6 | 2.15 | 57 | Fam20c | 39.47 | 19.44 | 2.03 |
| 58 | Eda2r | 14.0 | 6.6 | 2.16 | 58 | Adamts2 | 24.08 | 11.88 | 2.03 |
| 59 | Unkl | 12.9 | 6.1 | 2.13 | 59 | Glt28d2 | 18.70 | 9.24 | 2.02 |
| 60 | Slc40a1 | 29.4 | 14.1 | 2.10 | 60 | Entpd4 | 446.86 | 223.22 | 2.00 |
| 61 | Ece1 | 98.6 | 47.3 | 2.09 | | | | | |
| 62 | 2310061I04Rik | 26.0 | 12.6 | 2.07 | | | | | |
| 63 | H60b | 33.6 | 16.4 | 2.05 | | | | | |
| 64 | Ttll7 | 383.5 | 188.5 | 2.04 | | | | | |
| 65 | 1810011O10Rik | 10.9 | 5.4 | 2.05 | | | | | |
| 66 | Asb15 | 36.3 | 18.1 | 2.01 | | | | | |
| 67 | Nqo1 | 15.48 | 7.70 | 2.01 | | | | | |
| 68 | 0610009L18Rik | 13.98 | 6.96 | 2.01 | | | | | |

[a]Fold change (FC) was calculated after adding 0.01 to the original CPM values.

Expectedly, the male-enriched expression of PFKFB3 was detected at the protein level (Fig. 4f, Supplementary Fig. 5a).

The results above strongly suggested that male-predominant glycolysis can be achieved by the male-enriched expression of *Pfkfb3* alone. To verify this, we performed a knockdown study of the gene. We first examined if the gene could be suppressed by a general procedure for siRNA knockdown in cultured muscle fibers. These fibers were transfected with siRNAs for *Pfkfb3* or control siRNA for 6, 12, or 24 h, then the mRNA was quantified by qRT-PCR. The siRNA treatment suppressed *Pfkfb3* expression to approximately 25% at 6 h and then to 40–50% at 12 and 24 h after the treatment (Fig. 5a). Likewise, PFKFB3 was decreased at

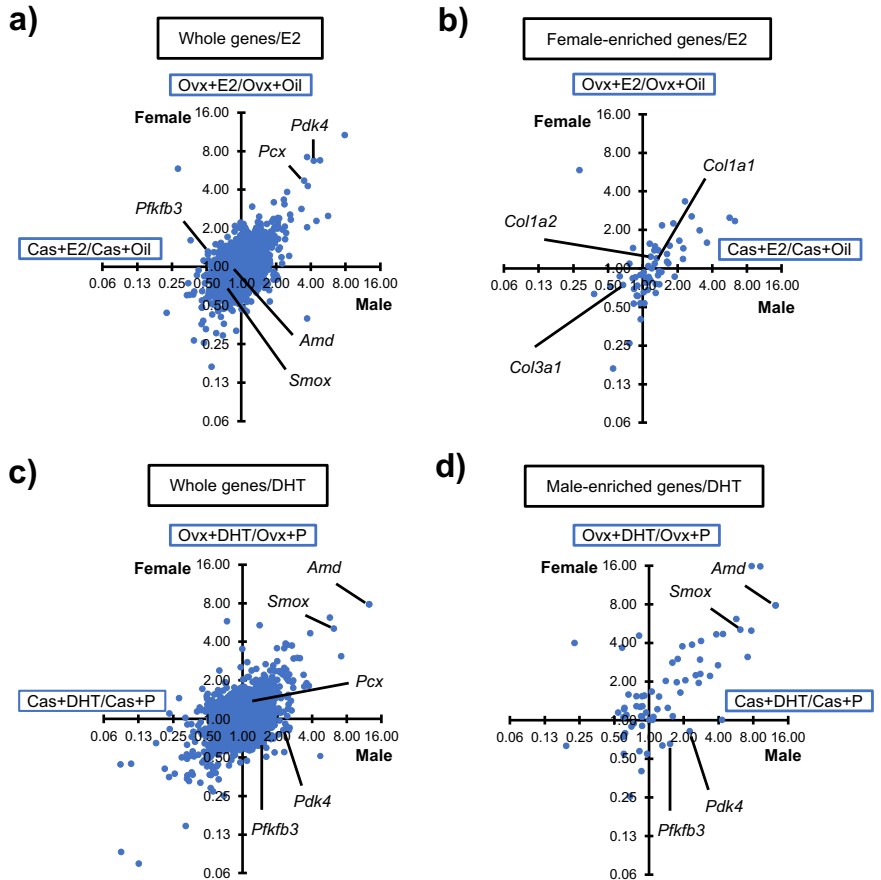

**Fig. 3 Identification of genes whose expression is potentially affected by sex steroids. a, b** Induction ratios by E2 treatment, specifically Cas+E2/Cas+O (horizontal axis) for males and Ovx+E2/Ovx+O (vertical axis) for females, were calculated and plotted for all expressed genes (**a**) and for the female-enriched genes (**b**). **c, d** Induction ratios by DHT treatment, specifically Cas+DHT/Cas+P for males (horizontal axis) and Ovx+DHT/Ovx+P for females (vertical axis), were plotted for all expressed genes (**c**) and for the male-enriched genes (**d**). The scales are logarithmic. The locations of *Pdk4*, *Pfkfb3*, *Pcx*, *Amd*, *Smox*, *Col1a1*, *Col1a2*, and *Col3a1*, are indicated.

the protein level by the si*Pfkfb3* treatment (Fig. 5b, Supplementary Fig. 5b). Investigation of the ECAR under the knockdown condition demonstrated that si*Pfkfb3* treatment downregulated the glycolytic activity in the male-derived muscle fibers at 12 h (Fig. 5c), with an even greater suppressive effect observed at 24 h. Of note, the activity of the si*Pfkfb3*-treated male-derived fibers decreased to approximately the same level as the siControl-treated female-derived fibers. Taken together, these results strongly suggest that the male-predominant glycolytic activity of quadriceps type IIB fibers can be established largely by the male-enriched expression of *Pfkfb3*.

**The possible contribution of PDK4 to female-predominant fatty acid metabolism.** Female skeletal muscles use fatty acid β-oxidation rather than glycolysis for energy production[12,39]. Therefore, we examined the expression of genes involved in fatty acid β-oxidation in the transcriptome datasets and using qRT-PCR (Fig. 6a, Supplementary Table 2c). Although many of these genes showed a tendency to be activated by E2, none demonstrated female-enriched or E2-enhanced expression over twice the baseline level. By contrast, our transcriptome data revealed E2-enhanced expression of *Pdk4*, which was further confirmed by qRT-PCR (Fig. 3, Fig. 6b, c). In addition to the level of the mRNA, PDK4 was increased in the E2-treated female muscle fibers at the level of protein (Fig. 6d, Supplementary Fig. 5c).

As for the function of PDK4, studies so far have established that the enzyme promotes the utilization of fatty acids for energy by suppressing the pyruvate dehydrogenase complex through phosphorylation[40] (Supplementary Fig. 4). Expectedly, the phosphorylation level of PDH (P-PDH) increased in the E2-treated female muscle (Fig. 6d, Supplementary Fig. 5c). Therefore, we hypothesized that female-predominant fatty acid β-oxidation is attributable to E2-induced *Pdk4*. To investigate this, we determined the fatty acid-dependent oxygen consumption rate (OCR) using cultured muscle fibers from male mice and diestrus female mice treated with or without E2 for 24 h. As shown in Fig. 6e, the fatty acid-dependent OCR was similar in male mice and diestrus female mice. As expected, the OCR was enhanced two-fold by E2 treatment.

Finally, we investigated the effect of *Pdk4* knockdown. When muscle fibers were treated with si*Pdk4*, the amount of *Pdk4* mRNA was decreased to 40% by 6-h treatment, and to 20% by 9- and 12-h treatments (Fig. 6f). This is consistent with the level of PDK4 protein decreased by the si*Pdk4* treatment (Fig. 6g, Supplementary Fig. 5d). Next, we examined whether the phosphorylation level of PDH is decreased by the si*Pdk4* treatment. As expected, it was found that the phosphorylation levels of PDH in the E2-treated female and male were significantly decreased (Fig. 6h, Supplementary Fig. 5e). Finally, muscle fibers treated with siRNA for 9-h were subjected to the aforementioned OCR assay. *Pdk4* knockdown resulted in a

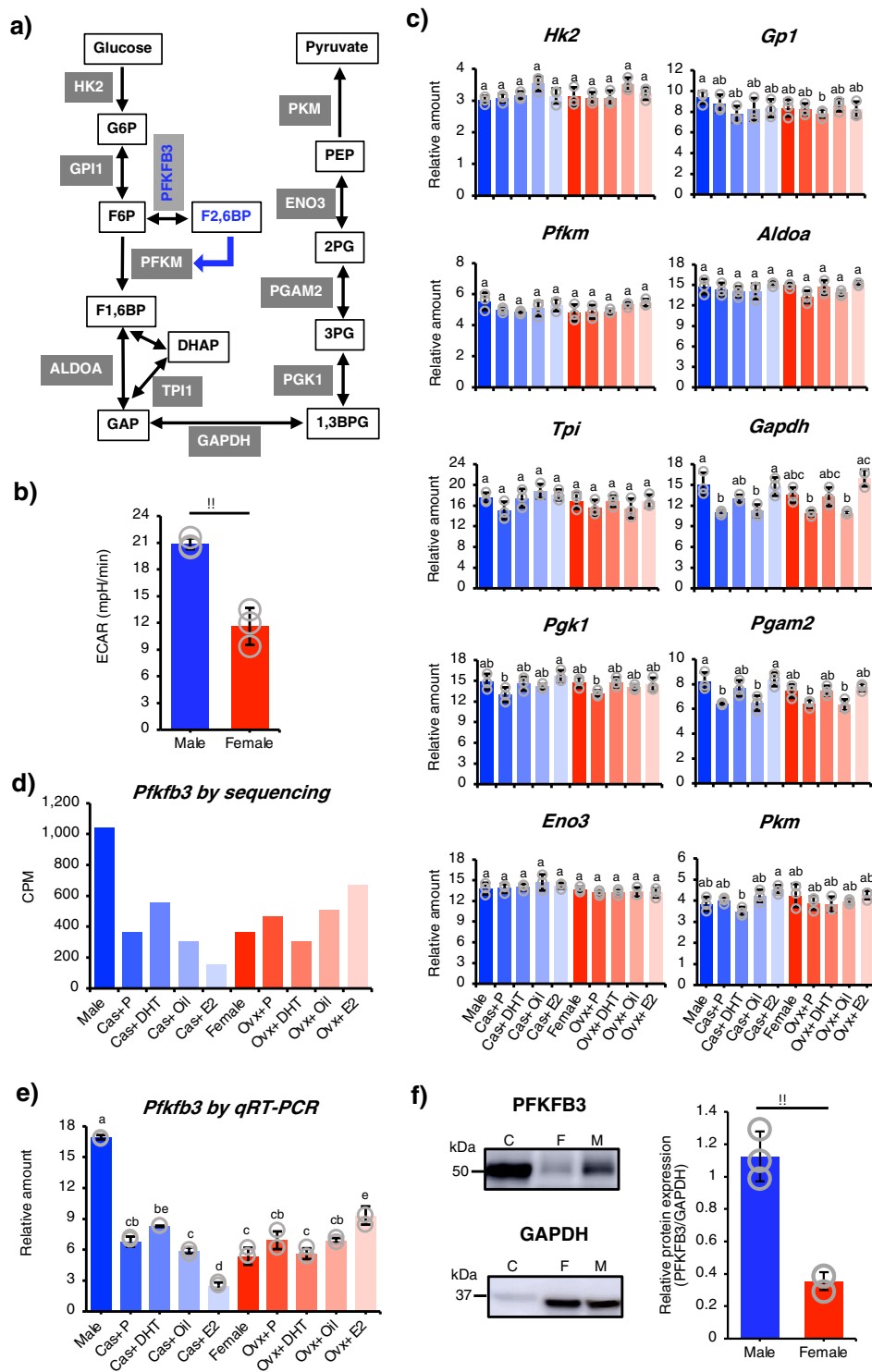

decrease of fatty acid-dependent OCR in all samples. Interestingly, the enhancement of fatty acid dependency by E2 treatment was nullified by *Pdk4* knockdown (Fig. 6i). Taken together, these results strongly suggest that female-predominant fatty acid utilization is attributable to E2-induced *Pdk4* gene expression.

## Discussion
Many transcriptome datasets have been obtained from skeletal muscles to evaluate the effects of exercise, metabolic diseases,

aging, etc. [41–43]. Some of them uncovered sexually dimorphic gene expression while others demonstrated the effects of sex steroids[14–17]. In addition to these transcriptome analyses, many studies have characterized the sexually dimorphic structures and functions of skeletal muscles[1]. One fundamental difference regards energy metabolism: male skeletal muscles preferentially utilize glycolysis, while female muscles tend to rely on mitochondrial fatty acid β-oxidation[7]. It was also shown, regardless of sex, that the main type of metabolism in some skeletal muscle fibers is anaerobic glycolysis, while in others it is aerobic

**Fig. 4 Male-biased glycolytic activity and *Pfkfb3* expression. a** The glycolytic pathway is illustrated, with enzymes shown in gray boxes and intermediate substances shown in open boxes. PFKFB3 mediates the reaction from F-6-P to F-2,6-BP, the latter of which acts as a strong activator of PFKM. **b** Male ($n = 3$) and female ($n = 3$) muscle fibers were prepared from type IIB-enriched regions of the quadriceps muscle (Supplementary Fig. 3a). The ECARs of cultured muscle fibers were examined, and the data were corrected by the ratio of the mean protein content in the fibers from the two sexes (male/female = 1.10) (Supplementary Fig. 3d). **$p < 0.01$. **c** The expressions of glycolytic genes were determined in the 10 experimental mouse groups (Supplementary Fig. 2) by qRT-PCR ($n = 3$ each group). Among paralogous genes, if any, the gene showing the highest expression was examined. **d** The expression of *Pfkfb3* was extracted from the transcriptome datasets. **e** The expression of *Pfkfb3* mRNA was determined by qRT-PCR ($n = 3$ each group). **f** The amounts of PFKFB3 protein were analyzed by western blotting. Whole proteins prepared from the type IIB-enriched areas of the quadriceps muscles of male (M) and female (F) were used. HeLa cell lysate was used as a control (C). Western blot images for PFKFB3 (upper left) and GAPDH (lower left) are shown. Full blot images are shown in Supplementary Fig. 5a. Three biologically independent samples were analyzed. The data were normalized to GAPDH and are presented as means ± SD (right). **$p < 0.01$. For **c** and **e**, the bars (means ± SD) with the same letter are not significantly different from each other ($p < 0.01$).

mitochondrial oxidation[44]. Therefore, the sexual dimorphism in the energy metabolism of skeletal muscle is due in part to the preponderance of glycolytic fibers in males and of oxidative fibers in females. However, it is also possible that sexually dimorphic metabolism is the result of differential metabolic activities intrinsic to male and female fibers. To investigate this issue, we focused on type IIB fibers, which are the most abundant type of fiber in fast-twitch muscles in rodents[45,46].

We found that male-predominant glycolytic activity could not be accounted for simply by enhanced glycolytic gene expression in males. Interestingly, however, *Pfkfb3* was one of the male-enriched genes identified in this study. PFKFB3 mediates the conversion of F-6-P (fructose-6-phosphate) to F-2,6-BP (fructose-2,6-bisphosphate), the latter of which acts as a potent allosteric activator of PFKM (a muscle type of PFK-1), one of the glycolytic rate-limiting enzymes[47]. During supramaximal exercise, activated glycolysis rapidly increases lactate concentrations, causing muscle fiber pH to become acidic, and simultaneously causes rapid reductions in oxygen and glucose concentrations[48]. These conditions are known to decrease glycolytic activity by suppressing the action of PFKM. Interestingly, however, suppression of the liver type of PFK-1 can be released by the robust action of F-2,6-BP produced by PFKFB3[49–51]. Because the concentration of F-2,6-BP was shown to correlate with the expression level of *Pfkfb3*/PFKFB3[52,53], we assumed that the male-enriched expression of *Pfkfb3* would ensure male-predominant glycolytic activity. Indeed, when knockdown was used to decrease the level of *Pfkfb3* gene expression in male-derived fibers to that observed in female-derived fibers, the glycolytic activity in the former decreased to levels similar to those in the latter. As described above, the male-predominant glycolytic activity of fast-twitch muscles is thought to be due to the larger number of type IIB fibers in male muscles. In addition, our study revealed for the first time that male type IIB fibers are intrinsically capable of driving glycolysis more robustly than female fibers through the male-biased expression of a single gene, *Pfkfb3*.

The results of this study suggest the potential importance of the sexually dimorphic expression of *Pfkfb3*. Studies so far have investigated the mechanism of *Pfkfb3* gene regulation from the viewpoint of glycolysis promotion in cancer cells. These studies implicated HIF1α (hypoxia-inducible factor 1α) in gene regulation[53,54]. In addition, testosterone and E2 were shown to activate *HIF1α* gene expression in prostate[55,56] and breast cancer cells[57], respectively, suggesting that sex steroids could induce *Pfkfb3* gene expression through *HIF1α* induction. Meanwhile, our current study of the quadriceps muscle suggested that still-unidentified factors besides testosterone are responsible for the male-enriched expression of *Pfkfb3*. Regarding these factors, it is interesting to note that the metabolic activities of preimplantation embryos are higher in males than in females[58], and the number of X chromosomes may contribute to sexually dimorphic

metabolism[59]. These studies suggest that genes localized on the sex chromosomes might play a role in the sexually dimorphic metabolism seen in XX and XY muscle fibers.

Our present study of cultured type IIB fibers demonstrated again the well-known fact that females preferentially utilize fatty acids for mitochondrial oxidation. This preference has been suggested to be due to E2-induced expression of genes related to fatty acid β-oxidation, including *Cpt1b* (which encodes a rate-limiting enzyme for β-oxidation), carnitine palmitoyltransferase[60,61], *Hadhb* (hydroxyacyl-CoA dehydrogenase), and *Pdk4* (pyruvate dehydrogenase kinase 4)[62]. Their results using whole gastrocnemius muscle correlate well with our findings using type IIB fibers, in that the induction ratio of *Pdk4* by E2 was more evident than those of other β-oxidation genes.

It has been established that PDK4 induces a metabolic shift from glycolysis to fatty acid β-oxidation through phosphorylation, thereby suppressing the pyruvate dehydrogenase complex[63]. Indeed, the level of *Pdk4* gene expression has been shown to correlate with the activity of fatty acid β-oxidation in cultured cells[40]. Taken together, we inferred that the preferential use of fatty acid β-oxidation in females is caused primarily by *Pdk4* gene expression induced by E2. Expectedly, E2 treatment enhanced fatty acid-dependent mitochondrial oxygen consumption in female-derived muscle fibers, and this enhancement was canceled by the knockdown of *Pdk4*. Although we cannot exclude the possibility that female-predominant fatty acid β-oxidation is attributable to E2-activated β-oxidation genes such as *Cpt1b*, our knockdown studies of cultured muscle fibers demonstrated that E2-induced *Pdk4* gene expression was a more likely cause.

We also observed E2-activated expression of the *Pcx* gene, whose product mediates an anaplerotic reaction to maintain tricarboxylic acid cycle flux by providing oxaloacetate. This reaction was reported to be critical for maintaining the oxidative function of mitochondria in skeletal muscle[64,65]. Therefore, *Pdk4* and *Pcx* may together coordinate female-predominant mitochondrial fatty acid β-oxidation through simultaneous induction by E2.

It has been accepted that cardiac muscle fibers of women have a higher activity of fatty acid β-oxidation than those of men[66,67]. This female-biased fatty acid β-oxidation was observed in the mice that developed a hypertrophied heart by exercise[68]. To comprehend the mechanism for the sexually dimorphic metabolism, transcriptomes were obtained from the cardiac muscles of both sexes[69–71]. A few genes required for fatty acid utilization were found as female-enriched genes. Unfortunately, however, none of the studies found *Pdk4* as the female-enriched gene, suggesting that distinct mechanism for female-biased fatty acid β-oxidation might work between the cardiac muscle and the type IIB fibers of skeletal muscle. Alternatively, because the estrus cycle was not considered in those studies, experiments to investigate the effects of E2 could uncover the implication of PDK4/*Pdk4* in female-biased fatty acid β-oxidation in the cardiac muscle.

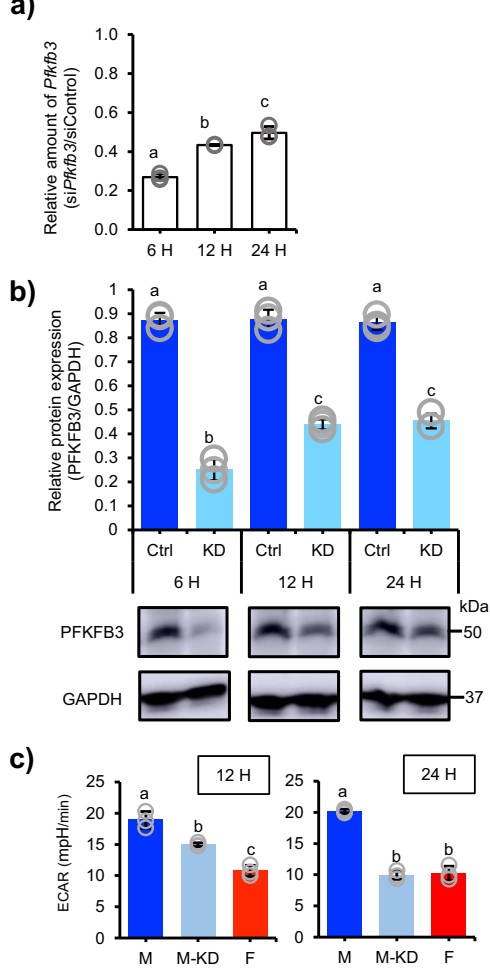

**Fig. 5 Impact of *Pfkfb3* knockdown on glycolysis in muscle fibers. a** Male quadriceps muscle fibers were transfected with si*Pfkfb3* (*n* = 3) or control siRNA (*n* = 3) for 6, 12, or 24 h. The amount of *Pfkfb3* mRNA was determined by qRT-PCR. The ratios of the amounts of *Pfkfb3* mRNA in the si*Pfkfb3*- and control siRNA-treated fibers are indicated. **b** The expressions of PFKFB3 and GAPDH were examined by western blotting at 6, 12, and 24 h after the siRNA transfection. The intensities of the PFKFB3 signals were normalized to those of GAPDH, and the relative amounts are presented as means ± SD (*n* = 3 in each group). Full blot images are shown in Supplementary Fig. 5b. **c** ECARs were examined in the male-derived fibers treated with control siRNA (M), male-derived fibers treated with si*Pfkfb3* (M-KD), and female-derived fibers treated with control siRNA (F). The fibers were treated with siRNA for 12 h (left) or 24 h (right). The data were corrected by the ratio of the mean protein content in the fibers from the two sexes. Data from three biologically independent samples were analyzed as described in the "Methods" section. The bars (means ± SD) with the same letter are not significantly different from each other (*p* < 0.01).

By focusing on gene expression in type IIB fibers in the quadriceps muscle, we unveiled the mechanisms of male-predominant glycolysis and female-predominant fatty acid β-oxidation. Interestingly, it appears that sexually dimorphic metabolism was achieved possibly by two genes, namely *Pdk4* and *Pfkfb3*, through transcriptional regulation by E2 and one or more unknown factor(s), respectively. Considering that skeletal muscle is the largest energy-consuming organ in the human body, our present findings may contribute to understanding the metabolism of both male and female individuals. In particular, our results may provide an insight into the metabolic properties of females,

whose E2 concentrations vary throughout life and during the estrus cycle. Our present study evaluated the characteristics of a single type of muscle fiber. However, since skeletal muscles consist of multiple fiber types[4], additional studies may provide deeper insights into a variety of functional differences, such as between fast and slow-twitch muscles, between the two sexes, and between healthy and pathological conditions.

## Methods

**Treatment of animals.** Male and female C57BL/6J mice (Japan SLC, Inc.) were gonadectomized or sham-operated at 3 weeks after birth. Three mice were kept in one cage and fed with standard CRF-1 chow (Oriental Yeast Co., Ltd., Tokyo, Japan). They had no interaction with individuals of opposite sexes. Treatment with DHT and E2 was performed as summarized in Supplementary Fig. 2. Every experimental group comprised three male and three female mice. Skeletal muscles (gastrocnemius, tibialis anterior, quadriceps, triceps, and soleus) were isolated at 8 weeks after birth and used for further studies. To determine estrus cycle phases in females, a vaginal smear test was conducted[72]. All animal experiment protocols were approved by the Animal Care and Use Committee of Kyushu University. All experiments were performed in accordance with the guidelines.

**mRNA sequencing and data processing.** The quadriceps muscles isolated from the aforementioned mice were immersed in RNAlater (Qiagen, Venlo, The Netherlands). Approximately 100 individual fibers were prepared from the muscles of 10 experimental groups (sham-operated male (*n* = 3) and female mice (*n* = 3) (CTR), pellet implantation male (*n* = 3) (Cas+P) and female mice (*n* = 3) (Ovx+P) after gonadectomy, oil injection male (*n* = 3) (Cas+Oil) and female mice (*n* = 3) (Ovx+Oil) after gonadectomy, DHT-treated male mice (*n* = 3) (Cas+DHT) and female mice (*n* = 3) (Ovx+DHT) after gonadectomy, and E2-treated male mice (*n* = 3) (Cas+E2) and female mice (*n* = 3) (Ovx+E2) after gonadectomy) (Supplementary Fig. 2) using fine forceps under a SMZ-U Zoom 1:10 stereomicroscope (Nikon, Tokyo, Japan)[73]. RNA was obtained individually from each fiber using TRIzol (Thermo Fisher Scientific, Waltham, MA, USA). cDNA was prepared from a small aliquot of each RNA and then subjected to PCR to distinguish fiber types using primer sets for myosin heavy chains (Supplementary Table 3). After the RNAs of type IIB fibers were collected, ribosomal RNA was removed using a NEBNext rRNA Depletion Kit (NEB, Ipswich, MA, USA). cDNA libraries for mRNA-seq were prepared using a NEBNext Ultra II RNA Directional Library Prep Kit (NEB). After the quality of the cDNA libraries was validated using an Agilent Bioanalyzer 2100 (Agilent Technologies, Santa Clara, CA, USA), the libraries were subjected to sequencing (NovaSeq 6000 System: Illumina, San Diego, CA, USA). STAR (version 2.7.3a)[74] and featureCounts (version 2.0.0)[75] were used for alignment and assembly of the sequence reads, respectively. *Mus musculus* genome assembly (mm10, NCBI) was used as the reference.

**qRT-PCR.** Three biologically independent samples were used for qRT-PCR using a CFX96 real-time PCR system (Bio-Rad, Hercules, CA, USA) and SYBR Select Master Mix (Thermo Fisher Scientific). The primer sets used are listed in Supplementary Table 3. The data were standardized using *Actb* (β-actin) and are presented as means ± standard deviation (SD). Statistical analysis was performed by one-way ANOVA followed by the post hoc Tukey HSD test[76] or the Student's *t*-test. Significant differences (*p* < 0.01) are indicated in the figures.

**Immunofluorescence analysis and CSA measurement.** Cryosections of the muscles of eight experimental groups (sham-operated male (*n* = 3) and female mice (*n* = 3) (CTR), gonadectomized mice (Cas for males (*n* = 3) and Ovx for females (*n* = 3)), DHT-treated male mice (*n* = 3) (Cas+DHT) and female mice (*n* = 3) (Ovx+DHT) after gonadectomy, and E2-treated male mice (*n* = 3) (Cas+E2) and female mice (*n* = 3) (Ovx+E2) after gonadectomy) (Supplementary Fig. 2) were subjected to immunofluorescence. Antibodies against MYH2B (myosin heavy chain type IIB) (1:1000), MYH2A (myosin heavy chain type IIA) (1:1000)[77], and laminin (1:1000) (Sigma-Aldrich, St. Louis, MO, USA) were used as the primary antibodies, while Mouse Anti-Rat IgG$_{2b}$-Alexa Fluor® 647 (1:500, SouthernBiotech, Birmingham, AL, USA), Mouse Anti-Rat IgG$_1$-Alexa Fluor 488® (1:500, SouthernBiotech), and Alexa Fluor 488-labeled Goat Anti-Rabbit IgG (1:500, Thermo Fisher Scientific) were used as the secondary antibodies. Nuclei were stained with 4′,6′-diamidino-2-phenylindole (DAPI) (Sigma-Aldrich). Fluorescence was observed using a LSM 700 confocal laser scanning microscope (Zeiss, Oberkochen, Germany). Histological images were analyzed using ImageJ software (Fiji)[78] to determine the CSAs of the muscle fibers. The average number of fibers analyzed in the sham-operated males and females, gonadectomized males (Cas) and females (Ovx), DHT-treated males (Cas+DHT) and females (Ovx+DHT) after gonadectomy, and E2-treated males (Cas+E2) and females (Ovx+E2) after gonadectomy are presented in corresponding order for each

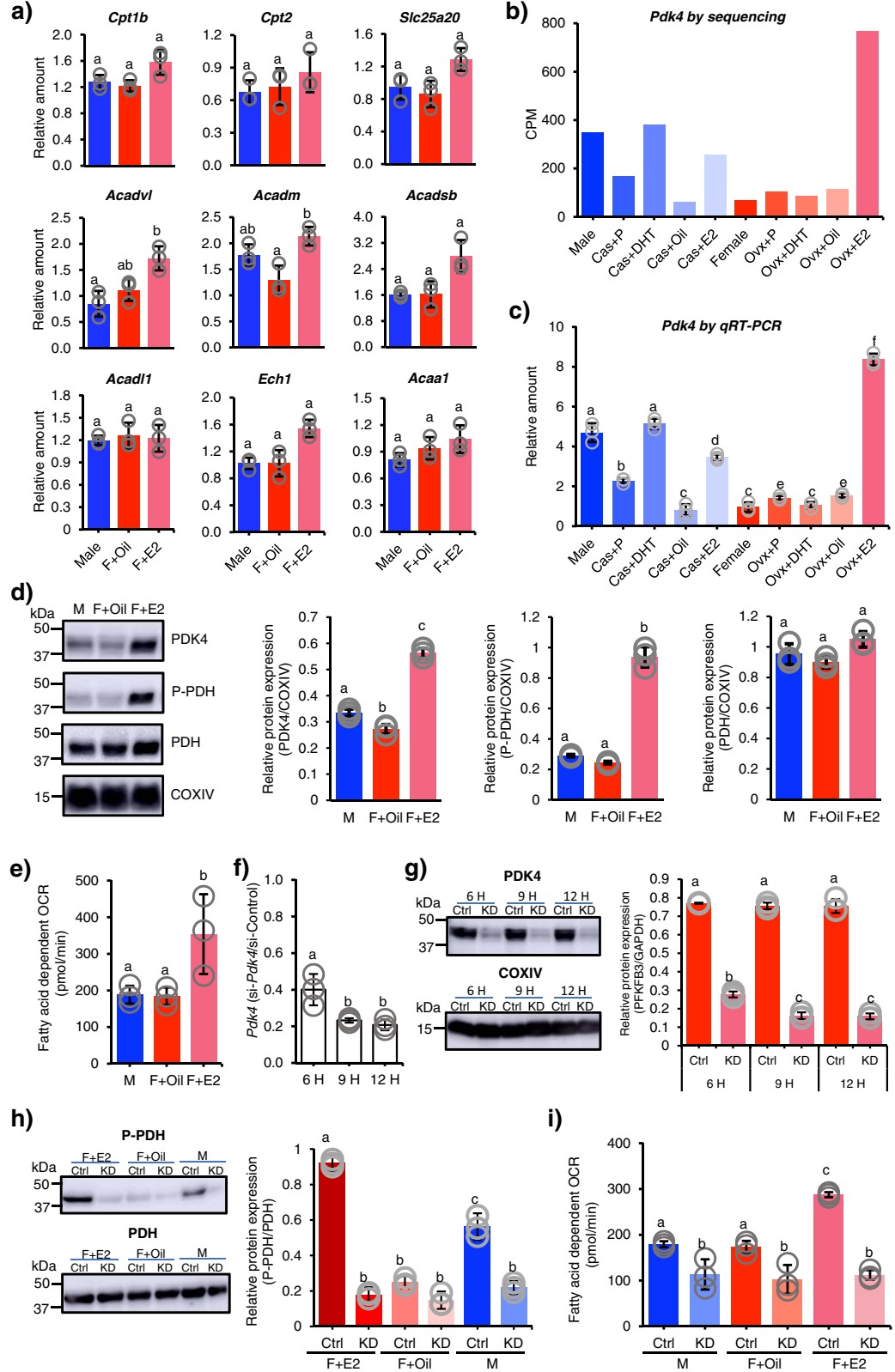

muscle as follows: for the quadriceps, 4800, 7500, 5900, 5100, 5500, 4900, 4900, and 6200 fibers, respectively; for the tibialis anterior, 2500, 2300, 3600, 3100, 4000, 3200, 3300, and 3300 fibers, respectively; for the gastrocnemius, 5900, 4600, 5700, 5400, 6200, 5100, 6700, and 4100 fibers, respectively; for the triceps, 4500, 4300, 4800, 4200, 5200, 4200, 4800, and 4300 fibers, respectively; and for the soleus, 50, 55, 65, 40, 65, 50, 63, 40 fibers, respectively. The CSAs are presented as means ± SD and were analyzed statistically by one-way ANOVA followed by the post hoc Tukey HSD test or the Student's t-test.

**Preparation of muscle fibers from the quadriceps muscle.** Type IIB fiber-enriched areas in quadriceps muscles were confirmed by an immunofluorescence study. In these regions, approximately >95% fibers were type IIB (Supplementary Fig. 3a). Living muscle fibers were prepared from the areas and cultured as described by Kitajima et al. [79]. In brief, the regions isolated from eight quadriceps muscles were incubated with 4 mg collagenase type I (Worthington Industries, Columbus, OH, USA) in 2 ml Dulbecco's Modified Eagle Medium (DMEM, Thermo Fisher Scientific) at 37 °C under 5% $CO_2$ for 1.5 h with gentle shaking by

**Fig. 6 Cancellation of fatty acid-dependent OCR by *Pdk4* knockdown. a** Muscle fibers were prepared from male ($n = 3$) (M) and diestrus female treated with oil ($n = 3$) (F + Oil) or E2 for 24 h ($n = 3$) (F + E2). The expression of genes implicated in fatty acid β-oxidation was examined by qRT-PCR. **b, c** Quadriceps type IIB fibers were prepared from the 10 experimental mouse groups (Supplementary Fig. 2). The expression profiles of *Pdk4* extracted from the sequence datasets (**b**) and determined by qRT-PCR ($n = 3$ each group) (**c**) are shown. **d** The muscle fibers used in **a** were used to determine the expressions of PDK4, P-PDH (phosphorylated PDH), and PDH by western blotting. The signals were semi-quantified as described in the "Methods" section. The intensities of the PDK4, P-PDH, and PDH signals were normalized to those of COXIV, and the relative amounts are presented as means ± SD ($n = 3$ each group). Full blot images are shown in Supplementary Fig. 5c. **e** The fatty acid-dependent OCRs of the fibers are shown ($n = 3$ each group). **f** The female muscle fibers were transfected with si*Pdk4* ($n = 3$) or control siRNA ($n = 3$) for 6, 9, or 12 h. The amount of *Pdk4* mRNA was determined by qRT-PCR. Ratios of the amounts of *Pdk4* mRNA between the si*Pdk4*- and control siRNA-treated fibers are indicated. **g** The effect of the knockdown was examined at the level of PDK4 protein. The intensities of the PDK4 signals were normalized to those of COXIV, and the relative amounts are presented as means ± SD ($n = 3$ each group). Full blot images are displayed in Supplementary Fig. 5d. **h** The levels of P-PDH and PDH in the muscle fibers were examined at 9 h after transfection with si*Pdk4* (KD) or control siRNA (Ctrl). Full blot images are displayed in Supplementary Fig. 5e. The intensities of the P-PDH were normalized to those of PDH signals, and the relative amounts are presented as means ± SD ($n = 3$ each group). **i** The fatty acid-dependent OCR was measured, and the data were corrected by the ratio of the mean protein content in the fibers from the two sexes. Data from three biologically independent samples were analyzed. The bars (means ± SD) with the same letter are not significantly different from each other ($p < 0.01$). Three biologically independent samples were used.

hand every 15 min. More than 80% of the recovered fibers were alive after overnight incubation. The viability of the isolated muscle fibers was assessed by microscopic observation. Healthy muscle fibers are long, translucent, and have clear surfaces without any shears, as described previously[80–82].

**Knockdown of *Pfkfb3* and *Pdk4* in cultured muscle fibers**. One hundred of the fibers prepared above were cultured on a plate coated with Matrigel matrix (Corning Incorporated, Corning, NY, USA) with 500 μl standard medium (DMEM containing 25 mM glucose supplemented with 20% fetal bovine serum (Thermo Fisher Scientific), 2% chick embryo extract (United States Biological, Salem, MA, USA), and 1% penicillin–streptomycin (PS, 10,000 U/ml, Thermo Fisher Scientific) at 37 °C under 5% $CO_2$ for 24 h. A mixture of two *Pfkfb3* siRNA duplexes (si*Pfkfb3*, 200 nM, SASI_Mm01_00034119 and SASI_Mm01_00034121 (Sigma-Aldrich)) or two *Pdk4* siRNA duplexes (si*Pdk4*, 100 nM, SASI_Mm01_00053023 and SASI_Mm01_00053024 (Sigma-Aldrich)) were transfected into the fibers using Lipofectamine RNAiMAX (Thermo Fisher Scientific) for 6, 9, 12, or 24 h in standard medium without PS. Stealth RNAi™ siRNA Negative Control, Med GC (Thermo Fisher Scientific) was used as a negative control. Transfection was performed according to the manufacturer's protocol and the procedure described by Huttner et al.[82]. RNAs were prepared from the fibers and then subjected to qRT-PCR of *Pfkfb3* and *Pdk4*.

**ECAR measurement**. ECAR was measured using a Seahorse XFe96 Analyzer (Agilent Technologies) basically according to the manufacturer's protocol. Fifteen muscle fibers were plated on a 96-well plate (Agilent Technologies) precoated with Matrigel matrix (Corning Incorporated). They were incubated with 200 μl of standard medium for 18 h, then cultured in XF base medium (Agilent Technologies) supplemented with 2 mM glutamine (Agilent Technologies) for 1 h at 37 °C without $CO_2$. Measurement of ECAR was started after the addition of 2 mM glucose.

To study the effect of *Pfkfb3* knockdown, muscle fibers were transfected with si*Pfkfb3* for 6 h after 18-h culture in a standard medium. After transfection, the fibers were further cultured in a standard medium for another 12 or 24 h. After 1-h culture in XF base medium, they were subjected to ECAR measurement. Approximately 10% of the fibers died during the transfection, and the numbers of living fibers varied among wells. Thus, the wells containing at least living 13 fibers by the end of the transfection were subjected to ECAR measurement. Since fiber sizes differed between males and females, the amount of protein in each fiber was determined (Supplementary Fig. 3d) and ECARs were corrected using the ratio of the mean protein content in the fibers of the two sexes (male/female = 1.10). Three biologically independent samples were used. Data are presented as means ± SD and were analyzed by one-way ANOVA followed by the post hoc Tukey HSD test or the Student's *t*-test.

**OCR measurement**. The OCR was measured using a Seahorse XFe96 Analyzer (Agilent Technologies) basically according to the manufacturer's protocol. Fifteen muscle fibers were prepared from each of the following groups: male mice ($n = 3$), oil-injected female mice in diestrus ($n = 3$), and female mice treated with E2 for 24 h ($n = 3$). All fibers were plated on a 96-well plate precoated with Matrigel matrix (Corning Incorporated), then incubated in 200 μl of standard medium for 18 h. Before OCR measurement, the medium was changed to XF base medium supplemented with 1 mM pyruvate, 2 mM glutamine, and 10 mM glucose, and the muscle fibers were incubated for 1 h without CO2. To determine the fatty acid-dependent OCR, 4 μM Etomoxir (Seahorse XF Mito Fuel Flex Test Kit, Agilent Technologies), an inhibitor of carnitine palmitoyl-transferase, was used. The degree to which the OCR was decreased by the inhibitor was defined as the fatty acid-dependent OCR.

To study the effect of *Pdk4* knockdown, fibers were transfected with si*Pdk4* or control siRNA after 18-h incubation in a standard medium. After the transfection, wells containing at least 13 living fibers were used for OCR measurement. The OCRs of fibers from males and females were corrected by the ratio of the mean protein content in the fibers of the two sexes (male/female = 1.10) (Supplementary Fig. 3d). Three biologically independent samples were used. Data are presented as means ± SD and were analyzed using one-way ANOVA followed by the post hoc Tukey HSD test.

**Western blotting**. Whole protein lysate was prepared from the type IIB fiber-enriched area in quadriceps muscles. The fibers were lysed using RIPA buffer (Sigma-Aldrich), followed by sonication (Branson Ultrasonics™ S-250A Model Sonifier™ Analog Cell Disrupter, Branson, Brookfield, CT, USA). Mitochondria were isolated from the muscle fibers as described by Garcia-Cazarin et al.[83] and lysed using RIPA buffer. The protein concentration was determined using BCA Protein Assay Kit (Thermo Fisher Scientific). 30 μg whole lysate or 10 μg mitochondrial proteins were subjected to SDS–polyacrylamide gel electrophoresis, followed by western blotting. Anti-PFKFB3 (1:2000, Proteintech, Rosemont, IL, USA), anti-GAPDH (1:10,000, Santa Cruz Biotechnology, Dallas, Texas, USA), anti-PDK4 (1:1000, Proteintech), anti-PDH (1:1000, Cell Signaling Technology, Danvers, MA, USA), anti-phospho-PDH α1 (1:1000, Cell Signaling Technology), and anti-COXIV antibodies (1:2000, Abcam, Cambridge) were used as primary antibodies. HRP-labeled anti-mouse IgG (Goat), (1:2000, Thermo Fisher Scientific) and HRP-linked F(ab')2 fragment of anti-Rabbit IgG, (Donkey) (1:2000. Cytiva, Marlborough, MA, USA) were used as secondary antibodies. Semi quantification of the proteins detected by western blotting was performed using ImageJ software (Fiji)[78]. Data (means ± SD) obtained from three biologically independent samples were analyzed by one-way ANOVA followed by the post hoc Tukey HSD test (Figs. 5b, 6d, g, and h) or the Student's *t*-test (Fig. 4f).

**Statistics and reproducibility**. Statistically significant differences between two groups were calculated using Student's *t*-test. Significant differences between multiple groups were calculated using one-way ANOVA followed by the post hoc Tukey HSD test. All experiments were performed with three biologically independent samples.

**Reporting summary**. Further information on research design is available in the Nature Research Reporting Summary linked to this article.

## Data availability
mRNA-seq data have been deposited in DDBJ under the accession code DRA010793 (https://ddbj.nig.ac.jp/DRASearch/). All source data underlying graphs in main figures are provided in Supplementary Data 1–5. All other data are available from the corresponding author on reasonable request.

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

## Acknowledgements

We thank Drs. Y. Imai and H. Sakai (Medical School, Ehime University) and Drs. H. Tanaka and H. Yamazaki (The Institute of Medical Science, The University of Tokyo) for their technical advice and discussion. We appreciate the technical assistance from The Research Support Center, Research Center for Human Disease Modeling, Kyushu University Graduate School of Medical Sciences. This work was supported by JSPS KAKENHI Grant Number JP20K08863 (T.B.), JP17H06427 (T.B., K.-I.M.), JP20H03436 (K.-I.M.), JP18H05527 (Ya.O.), JP19H05244 (Ya.O.), JP20H00456 (Ya.O.), JP20H04846 (Ya.O.), and JP20K21398 (Ya.O.); by JST CREST Grant Number JPMJCR16G1 (Ya.O.); by AMED under Grant Number JP20gk0210019 (K.-I.M.) and JP20ek0109489h0001 (Ya.O.).

## Author contributions

A.C., T.B., F.T., K.I., M.I., and K.-I.M. conceived and designed the experimental approaches and performed experiments and data analyses. A.C., T.B., and K.-I.M. prepared the manuscript. Ya.O. and M.S. contributed to the acquisition and computational analyses of the sequence data. Yu.O. contributed to the preparation of live muscle fibers.

## Competing interests

The authors declare no competing interests.
