## [Peer Review File · Communications Biology]

Reviewers' comments:

Reviewer #1 (Remarks to the Author):

In their manuscript entitled "Male-predominant glycolysis and female-predominant fatty acid utilization in muscles bestowed by sexually dimorphic expressions of Pfkfb3 and Pdk4", the authors analyzed transcriptomes obtained from quadriceps type IIB fibers. Two genes: Pfkfb3 and Pdk4 were differentially regulated between males and diestrus females, and were suggested to act as metabolic switches. Onadectomized, and sex steroid-treated mice of both sexes were used to verify findings in untreated mice.^[1]

In general, the study design is very robust and the manuscript contains some intriguing new findings that shed light on sex-differences in energy utilization between male and female mice. However, some of the results and their interpretations should be verified with alternative methods.

Specific comments:

The title of the manuscript should reflect that this study was done in rodents, with potentially limited translatability to humans. Suggested title: "Male-predominant glycolysis and female-predominant fatty acid utilization in rodent muscles are bestowed by sexually dimorphic expressions of Pfkfb3 and Pdk4"

The authors show that Pfkfb3 and Pdk4 are deregulated on the mRNA level, however no data are available for expression levels of these enzymes on the protein level. While there could be a correlation (and some of the physiological experiments suggest it), the authors should verify that changes seen on the mRNA level are also found on the protein level. In addition, the authors should also demonstrate that the siRNA experiments led to a reduction not only on the mRNA, but also on the protein level.

Data shown in Figure 5A (siRNA of Pfkfb3) seem miscalculated? The figure shows relative amount of siPfkfb3 treated cells over control siRNA treated cells. If siPfkfb3 treated cells reduce the amount of Pfkfb3 mRNA, there should be a reduction in the graph from 6-24 hours (as also seen in Figure 6E for siPdk4). However, the authors see an increase. Please correct or clarify.

Dietary phyto-hormones (e.g. phytoestrogens) are known to be able to modulate the gene program and metabolism in mammals. The authors should state which diet the mice received. Similarly, the authors should clarify how mice were housed before isolation of fibres and subsequent analysis using mRNA-seq/qRT-PCR. Testosterone levels in male mice differ significantly between male mice housed together, versus male mice that have interacted with estrus female mice (e.g. PMID: 24813700, 1162363).

What was the reasoning that the authors choose to analyze female mice during diestrus for their mRNA-seq experiments and not from female mice during (pro)estrus? Please explain.

It may be also worth noting in the discussion that more pronounced male-female gene expression differences may have been found if a mouse strain with higher levels of testosterone would have been used (see PMID: 15621042).^[1]

Supplemental table 1 (misspelled as 'Supplemntal table 1.') is in the the eyes of this reviewer more informative compared to Table 1. It may be worthwhile to move the current table 1 into the supplement and replace it with either all information presented in Suppl. table 1 or a subset (e.g. 'top 20 genes enriched in males and females')?

The authors state that "two key genes regulating energy metabolism, Pdk4 and Pcx (pyruvate carboxylase), were included (Fig. 3b)." However, Figure 3B does not depict these two enzymes. Maybe the sentence was attributing the wrong figure?^[1]

The figure legends state that "Significant differences ($p < 0.01$) are indicated by small letters." The authors should elaborate in the figure legend what letter denotes what comparison. Were all of the letters indicating p -values < 0.01 ?^[1]

PDK4 seems to only play minor roles in skeletal muscle of normal wildtype mice, as knockouts

display an unremarkable muscle phenotype, and the enzyme only gets properly induced upon starvation or diabetes/high-fat diet (e.g. PMIDs 16606348, 18430968). Although these studies only focused on male mice it is still questionable if slight differences observed between male/female in PDK4 may play such an important role as to warrant the statement that “sexually dimorphic metabolism was shown to be achieved PRIMARILY BY TWO genes, namely Pdk4 and Pfkfb3 [...]” (emphasis mine). Another question that comes to mind would be if differences in muscle metabolism/performance in female mice in estrus (presumably high PDK4) vs. diestrus (low PDK4), be (primarily) explainable by fluctuating PDK4 levels?

Figure 3 a,b: please indicate which ends of the axes correspond to Cas+E2, Cas+O for the horizontal axis, and OvX+E2, OvX+O for the vertical axis.

Figure 3 c,d: Please indicate which ends of the axes correspond to Cas+DHT, Cas+P for the horizontal axis, and OvX+DHT, OvX+P for the vertical axis.

The authors state that PFK-1 is “the glycolytic rate-limiting enzyme”. However, there is evidence that PFK1 is only be one of the rate-limiting enzymes/steps in glycolysis, which in addition may change depending on other factors (e.g. in cancer). Suggest to rephrase.

Reviewer #2 (Remarks to the Author):

In this manuscript, the authors demonstrated two key genes that function as switches between two sexually dimorphic metabolic pathways, and the results provide new insights into energy metabolism in the two sexes.

The research design was scientifically sound and the findings are of interest, but there are multiple concerns that need to be addressed prior to formal publication.

1. There is a lack of Western blotting analysis to verify the results from RT-PCR. It is hard to see whether or not these two genes were successfully silenced.
2. The pictures in figure 1-3 are well organized and nicely presented, but the layout of figure 4 is simple and poor.
3. Only knockdown test is not enough to prove the conversion action of pfkfb3 and pdk4 between the two pathways of energy metabolism in the two sexes. It will be better to perform a gain-of-function study, just like over-expressing pfkfb3 gene in female, or ectopically expressing pdk4 in male to parse the function as switches.
4. As shown in figure 5a, knockdown efficiency of pfkfb3 declined alongside time points. Was it plotted by mistake?
5. Line 229, it is better to replace “cancelled” by “nullified”.

Once the concerns above are fully addressed, the manuscript could be accepted for publication in this journal.

Reviewer #3 (Remarks to the Author):

The ultimate goal of this work was to gain novel insights into the molecular mechanisms underlying the sexually dimorphism of skeletal muscle. Whereas an extensive literature exists in regard to its morphological, biochemical and physiological sexual dimorphism, this work uniquely focused on the dimorphism at the level of the muscle fiber type and identified two key genes driving the differential metabolic activity of the type2B fibers: Pfkfb3 responsible for the male predominant glycolytic activity, and Pdk4 responsible for the female predominant fatty acid beta-oxidation. The novel findings described in this manuscript point to the existence of intrinsic

differential metabolic activities between male and female muscle fibers. This work is also well described, technically sound, and the data mostly support the conclusions. We encourage the authors to address a few concerns listed below.

1. Are there differences in the phosphorylation levels of PDH (as a measure of its enzymatic activity) in type2B fibers from males and females? It is indeed expected that the activity of this enzyme would be affected by the sexually dimorphic expression of Pdk4 and Pfkfb3. Similarly, it would have been important to show the phosphorylation levels of PDH in type2B fibers from males and females in the knockdown experiments.
2. The majority of the experiments rely on the isolation of individual fibers from the muscle, and most importantly on "live" ones. Indeed, the author stated that "More than 80% of the recovered fibers were alive after overnight incubation" (line 380-381). Could the authors clarify how viability was assessed?
3. Since it seems that the expression of Pfkfb3 is largely unaffected by DHT, what could the mechanism resulting in the male-enriched expression of Pfkfb3 be?
4. This Reviewer is curious as to whether the intrinsic differential metabolic activities identified in male and female muscle fibers may also be observed in the heart muscle. Is this perhaps worth to mention in the discussion?

We would like to thank all the reviewers for their insightful and helpful comments, suggestions, and discussions. Based on them, we have performed additional experiments and carefully revised our original manuscript. Our point-by-point responses to the reviewers' comments are shown below.

Reviewer #1

Comment 1,

The title of the manuscript should reflect that this study was done in rodents, with potentially limited translatability to humans. Suggested title: "Male-predominant glycolysis and female-predominant fatty acid utilization in rodent muscles are bestowed by sexually dimorphic expressions of Pfkfb3 and Pdk4"

Our response:

According to the reviewer's suggestion, we changed the title of the manuscript as follows: "Male-predominant glycolysis and female-predominant fatty acid utilization in rodent muscles are bestowed by sexually dimorphic expressions of *Pfkfb3* and *Pdk4*."

Comment 2,

The authors show that Pfkfb3 and Pdk4 are deregulated on the mRNA level, however no data are available for expression levels of these enzymes on the protein level. While there could be a correlation (and some of the physiological experiments suggest it), the authors should verify that changes seen on the mRNA level are also found on the protein level. In addition, the authors should also demonstrate that the siRNA experiments led to a reduction not only on the mRNA, but also on the protein level.

Our response:

As mentioned by the reviewer, our original manuscript contained quantitative data for *Pfkfb3* and *Pdk4* at the level of mRNA but not at the level of protein. According to the comment above, we examined whether the protein levels for PFKFB3 and PDK4 are consistent with their mRNA levels.

Western blotting with an anti-PFKFB3 antibody was performed to compare the protein levels between type IIB fibers of male (M) and female (F). Whole-cell lysate prepared from HeLa cells (C) was used as a positive control. Western blot images for PFKFB3 and GAPDH are shown on the next page. The signals detected by the analyses were semi-quantified using

ImageJ (Fiji) (Schindelin, 2012). The results indicated that the level of PFKFB3 protein was higher in male than female, while that of GAPDH was not sexually different. Three biologically independent samples were used. The amounts of PFKFB3 were normalized to GAPDH and are presented as means \pm SEM. The data (means \pm SEM) were analyzed by one-way ANOVA followed by the Student's t-test. The expression of PFKFB3 protein in the male fibers is significantly higher than that in the female ($p < 0.01$). As expected, this male-biased PFKFB3 protein level is consistent with the male-biased expression of its mRNA.

We have added these results in the revised manuscript as Fig. 4f and full blot images as Supplementary Fig. 5a. Accordingly, we have added descriptions in the result section (page 10, lines 195 - 196), in the methods section (page 22 line 459 - page 23 line 476), and in the legend to Fig. 4 (page 39 lines 762 - 767) in the revised manuscript.

We also examined whether *Pfkfb3* knockdown reduces PFKFB3 protein at the three time points, 6 H, 12 H, and 24 H, after siRNA treatment. PFKFB3 protein levels in the control siRNA- (Ctrl) and si*Pfkfb3*-treated (KD) male muscle fibers were examined by western blotting. As shown in the figure on the right, the protein level was decreased by the si*Pfkfb3* treatment. The expression of GAPDH was not affected by the treatment. Three biologically independent samples were used. The amounts of PFKFB3 were normalized to GAPDH and are presented as means \pm SEM. The data (means \pm SEM) were analyzed by one-way ANOVA followed by the post hoc Tukey HSD test. The expression of PFKFB3 protein in the si*Pfkfb3*-treated (KD) muscle fibers was significantly lower than that in the control siRNA-treated fibers (Ctrl) ($p < 0.01$) at all time points.

We have added these results as Fig. 5b and full blot images as Supplementary Fig. 5b in the revised manuscript. Accordingly, we have added descriptions in the result section (page 10 lines 203 - 204), in the methods section (page 22 line 459 - page 23 line 476), and in the legend to Fig. 5 (page 40 lines 773 - 776) in the revised manuscript.

Similarly, we performed western blotting to compare PDK4 protein levels between the fibers of the quadriceps muscles of male (M), oil-treated diestrus female (F+Oil), and estradiol (E2)-treated diestrus female (F+E2). As shown in the figure on the right,

the highest expression of PDK4 protein was found in the E2-treated female muscle fibers, which is consistent with the qRT-PCR result. Three biologically independent samples were used. The amounts of PDK4 were normalized to COXIV and are presented as means \pm SEM. The data (means \pm SEM) were analyzed by one-way ANOVA followed by the post hoc Tukey HSD test ($p < 0.01$).

We have added these results as Fig. 6d and full blot images as Supplementary Fig. 5c in the revised manuscript. Accordingly, we have added descriptions in the result section (page 11 lines 219 - 220), in the methods section (page 22 line 459 - page 23 line 476), and in the legend to Fig. 6 (page 42 lines 789 - 793) in the revised manuscript.

Next, we examined the effect of *Pdk4* knockdown on its protein level at 6 H, 9 H, and 12 H after transfection of *siPdk4*. As shown in the figure on the right, comparison between the samples transfected with control siRNA (Ctrl) and *siPdk4* (KD) demonstrated that PDK4

protein levels were decreased clearly by the *siPdk4* transfection. COXIV protein level was unaffected by the treatment. Three biologically independent samples were used. The amounts of PDK4 were normalized to COXIV and are presented as means \pm SEM. The data (means \pm SEM) were analyzed by one-way ANOVA followed by the post hoc Tukey HSD test ($p < 0.01$).

These results are included as Fig. 6g in the revised manuscript. Accordingly, we have added descriptions in the result section (page 11 lines 232 - 233), in the methods section (page 22 line 459 - page 23 line 476), and in the legend to Fig. 6 (page 42 lines 797 - 800). Full blot images have been included in the revised manuscript as Supplementary Fig. 5d.

Reference

Schindelin, J. *et al.* Fiji: An open-source platform for biological-image analysis. *Nat. Methods.* **9**, 676-682 (2012) doi:10.1038/nmeth.2019.

Comment 3,

Data shown in Figure 5A (siRNA of Pfkfb3) seem miscalculated? The figure shows relative amount of siPfkfb3 treated cells over control siRNA treated cells. If siPfkfb3 treated cells reduce the amount of Pfkfb3 mRNA, there should be a reduction in the graph from 6-24 hours (as also seen in Figure 6E for siPdk4). However, the authors see an increase. Please correct or clarify.

Our response:

We thank the reviewer for the comment above. As pointed out by the reviewer, we also thought that the knockdown results by siPfkfb3 were quite strange. Thus, we carefully performed the same study three times and found that the results were reproducible. Based on these studies, we reached a conclusion that the knockdown by the siRNA was most effective at 6 h after the treatment, and thereafter the amount of the mRNA increased. In addition to mRNA, the effect of the knockdown was examined at the level of protein as described in **our response to Comment 2** above. Consistent with the alteration of the mRNA, the amount of PFKFB3 was the lowest at 6 h and then increased at 12 and 24 h. Unfortunately, however, we currently have no idea about the reason. Further analyses such as investigation of Pfkfb3 gene regulation might give us cues to understand the exact reason for this phenomenon.

Comment 4,

Dietary phyto-hormones (e.g. phytoestrogens) are known to be able to modulate the gene program and metabolism in mammals. The authors should state which diet the mice received. Similarly, the authors should clarify how mice were housed before isolation of fibres and subsequent analysis using mRNA-seq/qRT-PCR. Testosterone levels in male mice differ significantly between male mice housed together, versus male mice that have interacted with estrus female mice (e.g. PMID: 24813700, 1162363).

Our response:

All mice used in this study were fed with a standard CRF-1 chow (Oriental Yeast Co., Ltd., Tokyo, Japan), which is widely used in animal facilities in our country. Although we could not obtain detailed information about the ingredients in CRF-1 from the company, we have not heard that the chow contains estrogenic substances at the level enough to affect experimental

outcomes. Information on the diet of the mice has been included in the Methods section of the revised manuscript (page 18 lines 344 - 345).

Three mice were group-housed, and they had no interaction with the individuals of opposite sexes until they were sacrificed. Moreover, it was reported that there was no significant difference in testosterone levels between male mice housed together for up to 12 weeks (Hohlbaum, 2020). Based on the report, our housing method does not seem to affect artificially the testosterone levels of the male mice used in our experiments. The housing conditions are described in the Method section of the revised manuscript (page 18 lines 344 - 346).

Reference

Hohlbaum, K. *et al.* Social enrichment by separated pair housing of male C57BL/6JRj mice. *Sci Rep.* **10**, 11165 (2020) doi:10.1038/s41598-020-67902-w.

Comment 5,

What was the reasoning that the authors choose to analyze female mice during diestrus for their mRNA-seq experiments and not from female mice during (pro)estrus? Please explain.

Our response:

When we started this study, we thought that the best plan to figure out changes of gene expression during estrus cycle is to obtain transcriptomes from every step of the cycle. However, it is not easy to catch precisely and reproducibly female mice at estrus and proestrus. Moreover, estradiol concentration in the blood rapidly changes at the steps and thus varies among individuals. By contrast, it is not difficult to catch the mice at diestrus. The estradiol concentration at the term is kept at low levels and does not vary among individuals. To avoid using female mice with different estradiol levels, we decided to use mice at diestrus.

This comment by the reviewer might be derived from the concern that not all E2-activated genes were caught by our study in which female-enriched genes were selected from the transcriptome of diestrus mice. To catch all E2-activated genes, transcriptomes from Ovx+oil and Ovx+E2 mice were compared. We believe that E2-activated genes were mostly recovered by this strategy.

Comment 6,

It may be also worth noting in the discussion that more pronounced male-female gene expression differences may have been found if a mouse strain with higher levels of testosterone would have been used (see PMID: 15621042).

Our response:

We thank the reviewer for the interesting comment. According to the suggestion by the reviewer, we have added the sentences below to raise a possibility that sexually different gene expression might be more pronounced if mice with higher testosterone levels were used in the study. The sentences below have been added in the result section of the revised manuscript (page 7 line 134 - page 8 line 137).

‘It was reported that testosterone levels were different among mouse strains, and that of C57BL/6 was significantly lower than those of CD-1, CH3, and FVB. Studies using the mice with higher testosterone levels might provide us with more pronounced sexually dimorphic gene expression.’

Comment 7,

Supplementary table 1 (misspelled as ‘Supplemntal table 1.’) is in the the eyes of this reviewer more informative compared to Table 1. It may be worthwhile to move the current table 1 into the supplement and replace it with either all information presented in Suppl. table 1 or a subset (e.g. ‘top 20 genes enriched in males and females’)?

Our response:

We thank the reviewer for the suggestion. We have replaced previous Table 1 with previous Supplementary Table 1 in the revised manuscript (page 43 line 808 -page 44 line 810). We also replaced previous Supplementary Table 1 with previous Table 1 in the revised Supplementary information (page 11 lines 103-109). Accordingly, we have changed the citation in the result section (page 7 line 114, page 8 line 142).

Comment 8,

The authors state that “two key genes regulating energy metabolism, Pdk4 and Pcx (pyruvate carboxylase), were included (Fig. 3b).” However, Figure 3B does not depict these two enzymes. Maybe the sentence was attributing the wrong figure?

Our response:

We carelessly made a mistake in citing the figure. We apologize for the error and thank the reviewer for pointing out our mistake. The figure that should be cited is figure 3a but not 3b. We have corrected the mistake in the revised manuscript (page 9 line 171).

Comment 9,

The figure legends state that “Significant differences ($p < 0.01$) are indicated by small letters.” The authors should elaborate in the figure legend what letter denotes what comparison. Were all of the letters indicating p -values < 0.01 ?

Our response:

We depicted these figures in reference to the studies by Piepho^{1,2} and Nanda *et al.*³ In this type of presentation, means (means \pm SEM in bar graphs) with the same letter are not significantly different from each other (p -values < 0.01). The letter-based symbols representing the presence or absence of significant differences are frequently used to show differences between groups. In this presentation, when bars are labelled with “a”, “ab” and “b”, we could conclude that there is a significant difference between the bars with “a” and “b”, but no significant difference between the bars with “a” and “ab”, and “b” and “ab”.

We have mentioned the meaning of the letter symbol in the legends to figures 1, 4, 5, and 6 in the revised manuscript (page 35 lines 728 - 729; page 39 lines 767 - 768; page 40 lines 781 - 782; and page 42 lines 806 - 807) by adding the sentence, ‘The box and whisker plots/The bars (means \pm SEM) with the same letter are not significantly different from each other ($p < 0.01$).’

References

1. Piepho, HP. An Algorithm for a letter-based representation of all-pairwise comparisons. *J Comput Graph Stat.* **13**, 456-466 (2004) doi:10.1198/1061860043515.
2. Piepho, HP. Letters in mean comparisons: What they do and don’t mean. *Agron J.* **110**, 431-434 (2018) doi:10.2134/agronj2017.10.0580.
3. Nanda, A., Mohapatra, B. B., Mahapatra, A. P. K., Mahapatra, A. P. K., & Mahapatra A. P. K. Multiple comparison test by Tukey’s honestly significant difference (HSD): Do the

confident level control type I error. *Int J Stat Appl Math.* **6**, 59-65 (2021)
doi:10.22271/math.2021.v6.i1a.636.

Comment 10,

PDK4 seems to only play minor roles in skeletal muscle of normal wildtype mice, as knockouts display an unremarkable muscle phenotype, and the enzyme only gets properly induced upon starvation or diabetes/high-fat diet (e.g. PMIDs 16606348, 18430968). Although these studies only focused on male mice it is still questionable if slight differences observed between male/female in PDK4 may play such an important role as to warrant the statement that “sexually dimorphic metabolism was shown to be achieved PRIMARILY BY TWO genes, namely Pdk4 and Pfkfb3 [...]” (emphasis mine). ?

Our response:

Reading the comment above by the reviewer, we recognized that we described intensely the contribution of PDK4 and PFKFB3 in the original manuscript. According to the suggestion by the reviewer, we have revised the original sentence as follow (page 16 lines 330 - 332).

ORIGINAL; Interestingly, sexually dimorphic metabolism was **shown to be** achieved **primarily** by two genes, namely *Pdk4* and *Pfkfb3*, through transcriptional regulation by E2 and one or more unknown factor(s), respectively.

REVISED; Interestingly, **it appears that** sexually dimorphic metabolism was achieved **possibly** by two genes, namely *Pdk4* and *Pfkfb3*, through transcriptional regulation by E2 and one or more unknown factor(s), respectively.

Comment 11,

Another question that comes to mind would be if differences in muscle metabolism/performance in female mice in estrus (presumably high PDK4) vs. diestrus (low PDK4), be (primarily) explainable by fluctuating PDK4 levels?

Our response:

We thank the reviewer for the interesting question. At least in our *in vitro* experiments, the type IIB fibers prepared from E2-treated female mice (high PDK4, mimics estrus) showed largely increased fatty acid utilization compared to those from diestrus females (Fig. 6i). This E2-activated fatty acid utilization disappeared from the *siPdk4*-treated fibers (Fig. 6i).

In addition to fatty acid utilization, we examined glucose-dependent and glutamine-dependent OCR (oxygen consumption rate). As shown in the results below, neither the glucose-dependent (right) nor

glutamine-dependent (left) OCR showed difference between the diestrus females and E2-treated females. Taken together, it is likely that

only fatty acid metabolism among energy metabolisms is fluctuating during estrus cycle and PDK4 is responsible for the fatty acid metabolism activated by E2. Therefore, it would be possible to conclude that the metabolic shift between estrus and diestrus is attributable to the E2-dependent PDK4 expression. We have not added these results in the revised manuscript. However, if the reviewer feels that these results are required, we are willing to add them.

We currently do not have any *in vivo* data showing that the muscle metabolism is different between diestrus and estrus of normally cycling females. We also have not examined yet whether muscle performance changes during estrus cycle. In addition, it is also interesting to see whether metabolic changes occur in other types of fibers other than type IIB. Moreover, studies with *Pdk4* knockout mice will support *in vitro* studies presented in this manuscript. As indicated by the reviewer, we still have remained many physiologically interesting questions unclear. Further studies are necessary to grasp the whole picture of sexually biased muscle metabolism and performance.

Comment 12,

Figure 3 a,b: please indicate which ends of the axes correspond to Cas+E2, Cas+O for the horizontal axis, and Ovx+E2, Ovx+O for the vertical axis. **Figure 3 c,d:** Please indicate which ends of the axes correspond to Cas+DHT, Cas+P for the horizontal axis, and Ovx+DHT, Ovx+P for the vertical axis. ?

Our response:

The horizontal and vertical axes of Fig. 3a and b indicate Cas+E2/Cas+Oil and Ovx+E2/Ovx+Oil, respectively, and those of Fig. 3c and d indicate Cas+DHT/Cas+pellet (P) and Ovx+DHT/Ovx+P, respectively. As shown below, we have labeled the horizontal and vertical axes of Fig. 3a and b with Cas+E2/Cas+Oil and Ovx+E2/Ovx+Oil, respectively, and the horizontal and vertical axes of Fig. 3c and d with Cas+DHT/Cas+P and Ovx+DHT/Ovx+P, respectively, in the revised manuscript.

Comment 13,

The authors state that PFK-1 is “the glycolytic rate-limiting enzyme”. However, there is evidence that PFK1 is only be one of the rate-limiting enzymes/steps in glycolysis, which in addition may change depending on other factors (e.g. in cancer). Suggest to rephrase.

Our response:

We thank the reviewer for letting us consider the point. We have rephrased the sentence as follows (page 13 lines 261 - 263).

ORIGINAL; the latter of which acts as a potent allosteric activator of PFKM (a muscle type of PFK-1), the glycolytic rate-limiting enzyme.

REVISED; the latter of which acts as a potent allosteric activator of PFKM (a muscle type of PFK-1), **one of** the glycolytic rate-limiting enzymes.

Reviewer #2

Comment 1,

There is a lack of Western blotting analysis to verify the results from RT-PCR. It is hard to see whether or not these two genes were successfully silenced.

Our response:

The same comment was given as ***Comment 2 by the reviewer 1***. Therefore, please refer to ***Our response*** to the ***Comment 2*** by the reviewer 1 above. In brief, we performed western blottings with the antibodies for PFKFB3 and PDK4. These studies demonstrated male biased expression of PFKFB3 and E2-activated expression of PDK4 (Fig. 4f and Fig. 6d in the revised manuscript, full blot images as Supplementary Fig. 5a and 5c), which are consistent with their expression in mRNA levels. Moreover, we confirmed the expression of PFKFB3 and PDK4 decreased by siRNA treatments (Fig. 5b and Fig. 6g in the revised manuscript, full blot images as Supplementary Fig. 5b and 5d).

Comment 2,

The pictures in figure 1-3 are well organized and nicely presented, but the layout of figure 4 is simple and poor.

Our response:

According to the suggestion by the reviewer, we have changed the layout of Fig. 4 in the revised manuscript. A new result for protein levels of PFKFB3 in the male and female muscles has been added as Fig. 4f in the revised Fig. 4.

Comment 3,

Only knockdown test is not enough to prove the conversion action of pfkfb3 and pdk4 between the two pathways of energy metabolism in the two sexes. It will be better to perform a gain-of-function study, just like over-expressing pfkfb3 gene in female, or ectopically expressing pdk4 in male to parse the function as switches.

Our response:

We agree with the notion that gain-of-function studies could strongly support our conclusion obtained from loss-of-function studies. Therefore, we attempted to overexpress PFKFB3 in cultured type IIB fibers using a lipofection reagent. Muscle fibers and HEK293

cells were cultured on 24-well plates and then transfected with ptd-Tomato-N1 plasmid, an expression vector for a red fluorescent protein, with Lipofectamine[®] 2000 (Thermo Fisher Scientific, Waltham, MA, USA). 0.8 or 1.6 μg plasmids were transfected with 1, 3, or 5 μl of the transfection reagent. As shown in the figures below, ptd-Tomato-N1 fluorescence was clearly observed in the HEK293 cells at 24, 48, and 96 h after the transfection. Unexpectedly, however, all our trials with the muscle fibers resulted in failure to detect ptd-Tomato-N1 fluorescence. Although we currently have no idea why the plasmid could not be transfected into the muscle fibers, the muscle fibers may have unique properties in terms of the nature of their membrane. In our impression, it will take longer time to establish the method to transfect plasmids efficiently into the muscle fibers. Although we understand the importance of the gain-of-function study, we cannot include them in our current manuscript. We hope that the gain-of-function study could support our present results after establishment of the method for transfection of plasmids into muscle fibers.

The results of our experiments are as follows:

HEK293

Muscle Fibers

Comment 4,

As shown in figure 5a, knockdown efficiency of pfkfb3 declined alongside time points. Was it plotted by mistake?

Our response:

We thank the reviewer for the comment. The same comment was given as *Comment 3 by the reviewer 1*. Therefore, please refer to *Our response* to the *Comment 3* above. In brief, we performed carefully same study three times, and found that the results were reproducible. In addition to mRNA, the effect of *Pfkfb3* knockdown was examined at the level of protein as described in *Our response to Comment 2 of the reviewer 1*. Consistent with the alteration of the mRNA, the amount of PFKFB3 was the lowest at 6 h and then increased slightly at 12 and 24 h. We have added these results as Fig. 5b and full blot images as Supplementary Fig. 5b in the revised manuscript. We currently have no idea about the reason for the unexpected effect of the *Pfkfb3* knockdown. Further analyses for *Pfkfb3* gene regulation and PFKFB3 protein degradation might give us cues to understand the exact reason for this phenomenon.

Comment 5,

Line 229, it is better to replace “cancelled” by “nullified”.

Our response:

We thank the reviewer for the kind suggestion. We have replaced ‘cancelled’ by ‘nullified’ in the revised manuscript (page 12 line 238).

Reviewer #3

Comment 1,

Are there differences in the phosphorylation levels of PDH (as a measure of its enzymatic activity) in type2B fibers from males and females? It is indeed expected that the activity of this enzyme would be affected by the sexually dimorphic expression of Pdk4 and Pfkfb3. Similarly, it would have been important to show the phosphorylation levels of PDH in type2B fibers from males and females in the knockdown experiments.

Our response:

This comment by the reviewer partially overlaps with *the comment 2 by the reviewer 1* and *the comment 1 by the reviewer 2*. According to the suggestion by the reviewer, we compared phosphorylation levels of PDH (P-PDH) together with expression levels of PDH and PDK4 in the quadriceps type IIB fibers by western blot analyses. COXIV was used as a control. The signals detected by the analyses were semi-quantified using ImageJ (Fiji) (Schindelin, 2012).

As shown in the figures below, the proteins indicated were detected in the muscle fibers from male (M), diestrus female treated with oil (F+Oil), and diestrus female treated with estradiol (F+E2) for 24 h. The results of the western blotting are shown at the left. The signals were semi-quantified as described in the Methods. Three biologically independent samples were used. The amounts of PDK4, P-PDH, and PDH were normalized to COXIV and are presented as means \pm SEM. The data (means \pm SEM) were analyzed by one-way ANOVA followed by the post hoc Tukey HSD test ($p < 0.01$).

The expression level of PDK4 (the second figure from the left) was slightly higher in male than in female, and that of female was increased more than 2-fold by the E2 treatment, which is consistent with the level of *Pdk4* mRNA. As expected, the level of P-PDH was higher in the E2-treated female than in the oil-treated female (the second figure from the right) whereas the level of PDH was comparable among the three samples (the right figure). These results

suggested that E2 induced PDK4 protein expression and thereby induced phosphorylation of PDH. We have included the results above as Fig. 6d in the revised manuscript (full images of western blotting were shown as Supplementary Fig. 5c). Accordingly, we have added a description in the text (page 11 lines 223 - 224), in the methods section (page 22 line 459 - page 23 line 476), and in the figure legend (page 42 lines 789 - 793).

We examined whether the P-PDH is decreased by *Pdk4* knockdown. As shown in the figures on the right, the level of P-PDH was decreased significantly in the E2-treated female fibers and male fibers by *Pdk4* knockdown.

We have added these results in the revised manuscript as Fig. 6h (full blot images were included as Supplementary Fig. 5e). Accordingly, we have added a description in the text (page 11 line 233 - page 12 line 235), in the methods section (page 22 line 459 - page 23 line 476), and in the figure legend (page 42 lines 800 - 803).

Reference

Schindelin, J. *et al.* Fiji: An open-source platform for biological-image analysis. *Nat. Methods.* **9**, 676-682 (2012) doi:10.1038/nmeth.2019.

Comment 2,

The majority of the experiments rely on the isolation of individual fibers from the muscle, and most importantly on “live” ones. Indeed, the author stated that “More than 80% of the recovered fibers were alive after overnight incubation” (line 380-381). Could the authors clarify how viability was assessed?

Our response:

The viability of the isolated muscle fibers was assessed by microscopic observation according to the previously established criteria^{1,2,3}. As shown in the figure on the next page, unhealthy muscle fibers are shrunk (red arrows). By contrast, healthy fibers are long, translucent, and have clear surfaces without any shears.

We have added a description of viability criteria in the methods section of the revised manuscript (page 20 lines 401 - 403).

References

1. Pasut, A., Jones, A. E. & Rudnicki, M. A. Isolation and culture of individual myofibers and their satellite cells from adult skeletal muscle. *J Vis Exp.* **73**, e50074 (2013) doi:10.3791/50074.
2. Gallot, Y. S., Hindi, S. M., Mann, A. K. & Kumar, A. Isolation, culture, and staining of single myofibers. *Bio-protocol.* **6**, e1942 (2016) doi:10.21769/BioProtoc.1942.
3. Hüttner, S. S. *et al.* Isolation and culture of individual myofibers and their adjacent muscle stem cells from aged and adult skeletal muscle. *Methods Mol Biol.* **2045**, 25-36 (2019). doi:10.1007/7651_2019_209.

Comment 3,

Since it seems that the expression of Pfkfb3 is largely unaffected by DHT, what could the mechanism resulting in the male-enriched expression of Pfkfb3 be?

Our response:

We thank the reviewer for this comment. As mentioned by the reviewer, it is unlikely that DHT (testosterone) is the major factor to activate *Pfkfb3* gene expression, although castration decreased the gene expression. These results suggested that testicular factors other than testosterone could be responsible for the male-enriched expression of *Pfkfb3*. Honestly, we currently do not have any plausible candidate factor to activate the gene expression.

Comment 4,

This Reviewer is curious as to whether the intrinsic differential metabolic activities identified in male and female muscle fibers may also be observed in the heart muscle. Is this perhaps worth to mention in the discussion?

Our response:

We thank the reviewer for the suggestion. As mentioned by the reviewer, it has been accepted that the cardiac muscle shows sexually dimorphic metabolism. The sentences below have been added to the revised manuscript (page 16 lines 317 - 327).

‘It has been accepted that cardiac muscle fibers of women have a higher activity of fatty acid β -oxidation than those of men^{1,2}. This female-biased fatty acid β -oxidation was observed in the mice that developed a hypertrophied heart by exercise³. To comprehend the mechanism for the sexually dimorphic metabolism, transcriptomes were obtained from the cardiac muscles of both sexes^{4,5,6}. A few genes required for fatty acid utilization were found as female-enriched genes. Unfortunately, however, none of the studies found *Pdk4* as the female-enriched gene, suggesting that distinct mechanism for female-biased fatty acid β -oxidation might work between the cardiac muscle and the type IIB fibers of skeletal muscle. Alternatively, because estrus cycle was not considered in those studies, experiments to investigate the effects of E2 could uncover implication of PDK4/*Pdk4* in female-biased fatty acid β -oxidation in the cardiac muscle.’

References

1. Kadkhodayan, A. *et al.* Sex affects myocardial blood flow and fatty acid substrate metabolism in humans with nonischemic heart failure. *J Nucl Cardiol.* 24, 1226-1235 (2017). doi: 10.1007/s12350-016-0467-6.
2. Ventura-Clapier, R. *et al.* Sex in basic research: concepts in the cardiovascular field. *Cardiovasc Res.* **113**, 711-724 (2017). doi: 10.1093/cvr/cvx066.
3. Foryst-Ludwig, A. *et al.* Sex differences in physiological cardiac hypertrophy are associated with exercise-mediated changes in energy substrate availability. *Am J Physiol Heart Circ Physiol.* **301**, 115-122 (2011) doi:10.1152/ajpheart.01222.2010.

4. Trexler, C. L., Odell, A. T., Jeong, M. Y., Dowell, R. D., & Leinwand, L. A. Transcriptome and functional profile of cardiac myocytes is influenced by biological sex. *Circ Cardiovasc Genet.* **10**, e001770 (2017). doi: 10.1161/circgenetics.117.001770.
5. Synnergren, J. *et al.* Transcriptional sex and regional differences in paired human atrial and ventricular cardiac biopsies collected in vivo. *Physiol Genomics.* **52**, 110-120 (2020). doi: 10.1152/physiolgenomics.00036.
6. Camila, M. *et al.* Sex differences in gene expression and regulatory networks across 29 human tissues. *Cell Rep.* **31**, 107795 (2020). doi: 10.1016/j.celrep.2020.107795.

REVIEWERS' COMMENTS:

Reviewer #3 (Remarks to the Author):

The authors have fully addressed the concerns I raised in the previous round of review. I do not have any additional comments.